# Distribution of Antimicrobial Resistance and Biofilm Production Genes in the Genomic Sequences of *S. aureus*: A Global In Silico Analysis

**DOI:** 10.3390/antibiotics14040364

**Published:** 2025-04-01

**Authors:** Ana Carolina Silva-de-Jesus, Rafaela G. Ferrari, Pedro Panzenhagen, Anamaria M. P. dos Santos, Ana Beatriz Portes, Carlos Adam Conte-Junior

**Affiliations:** 1Center for Food Analysis (NAL), Technological Development Support Laboratory (LADETEC), Federal University of Rio de Janeiro (UFRJ), Cidade Universitária, Rio de Janeiro 21941-853, Brazil; carolana08@ufrj.br (A.C.S.-d.-J.); rafaelaferrari@yahoo.com.br (R.G.F.); pedropanzen@iq.ufrj.br (P.P.); anamariasantos@id.uff.br (A.M.P.d.S.); aportes@id.uff.br (A.B.P.); 2Laboratory of Advanced Analysis in Biochemistry and Molecular Biology (LAABBM), Department of Biochemistry, Federal University of Rio de Janeiro (UFRJ), Cidade Universitária, Rio de Janeiro 21941-853, Brazil; 3Graduate Technology Biotechnology, Universidade Estadual do Rio de Janeiro Campus-ZO, Manuel Caldeira de Alvarenga, Rio de Janeiro 23070-200, Brazil; 4Graduate Program in Food Science (PPGCAL), Institute of Chemistry (IQ), Federal University of Rio de Janeiro (UFRJ), Cidade Universitária, Rio de Janeiro 21941-909, Brazil; 5Graduate Program in Veterinary Hygiene (PPGHV), Faculty of Veterinary Medicine, Fluminense Federal University (UFF), Vital Brazil Filho, Niterói 24220-000, Brazil; 6Laboratory of Microorganism Structure, Department of General Microbiology, Institute of Microbiology Paulo de Góes (IMPG), Federal University of Rio de Janeiro (UFRJ), Rio de Janeiro 21941-853, Brazil; 7Molecular & Analytical Laboratory Center, Department of Food Technology, Faculty of Veterinary, Universidade Federal Fluminense, Niterói 24220-900, Brazil

**Keywords:** *Staphylococcus aureus*, antimicrobial resistance (AMR), biofilm production, mobile genetic elements (MGEs), whole genome sequencing (WGS)

## Abstract

**Background:** *Staphylococcus aureus* constitutes a significant public health threat due to its exceptional adaptability, antimicrobial resistance (AMR), and capacity to form biofilms, all of which facilitate its persistence in clinical and environmental settings. **Methods:** This study undertook an extensive in silico analysis of 44,069 *S. aureus* genomic sequences acquired from the NCBI database to assess the global distribution of biofilm-associated and resistance-associated genes. The genomes were categorized into human clinical and environmental groups, with clinical samples representing a predominant 96%. **Results:** The analysis revealed notable regional discrepancies in sequencing efforts, with Europe and North America contributing 76% of the genomes. Key findings include the high prevalence of the *ica* locus, which is associated with biofilm formation, and its robust correlation with other genes, such as *sasG*, which was exclusively linked to SCC*mec* type IIa. The AMR gene analysis revealed substantial genetic diversity within environmental samples, with genes like vga(E) and erm being identified as particularly prominent. The clonal complex analysis revealed ST8 (USA300) and ST5 as the predominant types in human clinical isolates, while ST398 and ST59 were most frequently observed in environmental isolates. SCC*mec* type IV was globally prevalent, with subtype Iva being strongly associated with ST8 in North America and subtype IVh with ST239 in Europe. **Conclusions:** These findings underscore the dynamic evolution of *S. aureus* via mobile genetic elements and highlight the necessity for standardized metadata in public genomic databases to improve surveillance efforts. Furthermore, they reinforce the critical need for a One Health approach in monitoring *S. aureus* evolution, particularly concerning the co-dissemination of biofilm and resistance genes across various ecological niches.

## 1. Introduction

### 1.1. Resistance and Dissemination Potential

Methicillin-resistant *Staphylococcus aureus* (MRSA) arises from the expression of the *mec*A gene, which encodes a penicillin-binding protein (PBP2A) exhibiting reduced affinity for β-lactam antibiotics [1]. The *mec*A gene is located within a mobile genetic element known as the staphylococcal chromosomal cassette (SCC*mec*) [2]. SCC*mec* is bordered by cassette chromosome recombinase genes (*ccrA*/*ccrB* or *ccrC*), facilitating both the intra- and interspecies horizontal gene transfer of SCC*mec* [3]. The remaining regions of SCC*mec* consist of non-essential components, including additional resistance genes to metals and antibiotics, carried by transposons and plasmids, as well as genes of unknown function [4]. These regions are referred to as J regions (Figure 1).

Thirteen distinct structural types of SCC*mec* have been documented in *S. aureus*, with sizes ranging from 20 to 70 kb. These types correspond to combinations of *mec* complex classes (A–E), which are determined by the presence or absence of regulatory genes, insertion sequences, and ccr allotypes [3,5]. While the SCC*mec* is the most extensively studied element due to its role in antibiotic resistance, several pseudo-SCC elements, which do not harbor the *mec* complex or *ccr* complex, have also been identified via the orfX (open reading frame X) site [1]. The *orfX* region (also known as *rmlH*, which encodes an rRNA methyltransferase) in *Staphylococcus aureus* plays a fundamental role in the species’ genomic plasticity, serving as the primary insertion site for SCC-type mobile genetic elements, including SCC*mec* cassettes responsible for methicillin resistance. Studies have shown that *orfX* is highly conserved, but its genomic neighborhood can vary significantly due to the acquisition and recombination of different SCC types and their remnants (SCC-like elements) [6].

The presence of SCC*mec* and SCC-like remnants suggests previous integration and excision events, highlighting the dynamic nature of genetic mobilization in *S. aureus*. These fragments may contain functional genes or pseudogenes that, while not directly conferring resistance, serve as genetic reservoirs and contribute to the bacterium’s genomic plasticity [7,8]. For instance, the methicillin-sensitive *S. aureus* (MSSA) strain ATCC 25923 has been found to carry a 5877 bp fragment inserted at the *attB* SCC site downstream of *orfX* [7]. This fragment exhibits structural features similar to SCC*mec* at both ends—specifically, incomplete inverted repeats and 15 bp direct repeats—but lacks drug resistance genes or *ccr* genes [8].

Moreover, partial excision of SCC*mec* may result in the retention of regulatory or resistance genes that are not conventionally associated with SCC*mec*, contributing to the phenotypic diversity and the environmental adaptation of the pathogen [9]. Variability in *orfX*-associated SCC elements may influence not only antimicrobial resistance but also the virulence and persistence of *S. aureus* strains. Several studies indicate that atypical SCC*mec* variants, containing hybrid or vestigial elements, may alter gene expression and affect pathogen–host interactions. This underscores the importance of detailed analyses of these remnants and their potential role in pathogen evolution [9,10].

Notably, the absence of a complete SCC*mec* in certain lineages does not necessarily indicate the absence of genetic elements associated with resistance, reinforcing the need for a genome-wide approach to characterize the genetic plasticity of *S. aureus* [11]. These elements—whether intact or fragmented—may carry genes that enhance staphylococcal survival and virulence, including those conferring resistance to heavy metals, influencing cell wall biosynthesis, and encoding additional resistance determinants [6,7]. The original reservoir of SCC*mec* in *S. aureus* remains uncertain, though it may have originated in coagulase-negative staphylococcal species [7,8].

These elements may carry a variety of genes that contribute to staphylococcal survival and virulence, including those conferring resistance to heavy metals, influencing cell wall biosynthesis, and encoding additional resistance determinants [2,5,12]. The original reservoir for SCC*mec* in *S. aureus* remains uncertain but may have originated in coagulase-negative staphylococcal species [13,14,15].

Humans serve as asymptomatic carriers of *S. aureus*, which harbors antimicrobial resistance genes in the nasal passages, throat, and skin [16]. As a result, food handling by industry personnel has been identified as a potential vector for foodborne contamination [17]. In hospital environments, contamination of medical devices through improper handling contributes to severe infections, particularly in immunocompromised patients [18]. Furthermore, the ability of *S. aureus* to form biofilms enables it to endure hostile environments, such as metal surfaces in the food industry, plastic coatings, and both human and veterinary medical devices, thus heightening the risk of recurrent contamination in both animal and human populations [19].

### 1.2. Biofilm Architecture and Composition Variability

Biofilm is a bacterial community composed of diverse extracellular components, including polysaccharides, proteins, and teichoic acid. The formation of a bacterial biofilm is a multifactorial and intricate process, typically delineated into three phases involving distinct molecular mechanisms: attachment, accumulation/maturation, and detachment/dispersion [20,21,22,23]. The molecular foundations of *S. aureus* biofilm formation remain incompletely understood due to its structural complexity and dependence on a variety of external factors [22]. However, certain elements have been identified, such as the extracellular matrix in clinical *S. aureus* biofilms, which consists of cytoplasmic proteins being linked to the cell surface in response to lowered pH levels [24]. Additionally, factors like glucose and NaCl availability can modulate cellular hydrophobicity and trigger biofilm formation in both food and clinical strains, which exhibit more pronounced biofilm phenotypes than strains not exposed to such conditions [24,25].

*S. aureus* produces at least two types of biofilms: the *ica*-dependent type, mediated by the intercellular adhesive polysaccharide (PIA), also known as polysaccharide-N-acetyl-1,6-β-glucosamine (PNAG), and the *ica*-independent type [26]. The genes within the *ica* locus are responsible for the biosynthesis of PIA [26,27]. The *ica*-independent biofilms, on the other hand, are primarily composed of proteins containing the LPXTG motif (where X represents any amino acid) [22,28]. This extensive group of LPXTG-motif proteins in *S. aureus* is encoded by a variety of genes, including *fnbpA/fnbpB*, which code for fibronectin-binding proteins A and B [29]; *clfA/clfB*, associated with fibrinogen-binding protein aggregation factors A and B [30]; *bap*, linked to biofilm-associated protein (Bap) [31]; *sdr*, encoding proteins of the serine-aspartate repeat (Sdr) family [32]; *ebpS*, which encodes elastin-binding protein (Ebps) [33]; and *cna*, associated with collagen-binding protein (Cna) [23], among others.

The relative significance of each biofilm component appears to be strain-specific and influenced by growth conditions and environmental factors. Moreover, clones derived from different sources, whether food-related or infection-related, may exhibit differential biofilm-forming capacities [25,32]. For instance, some studies report a higher prevalence of exopolysaccharides (EPSs) in strains isolated from food, particularly poultry [34]. In contrast, strains derived from human infections tend to rely more heavily on cell wall-anchored proteins (CWAs) [35].

Another important consideration is the unclear role of all MSCRAMMs (microbial surface components recognizing adhesive matrix molecules) in the biofilm formation process [24]. Bap, which shows structural similarities to surface proteins in both Gram-negative bacteria (e.g., *Pseudomonas aeruginosa* and *Salmonella enterica* serovar Typhi) and Gram-positive bacteria (e.g., *Enterococcus faecalis*), was the first protein identified as crucial for biofilm formation in staphylococcal strains isolated from the mammary glands of ruminants with mastitis [25]. Although *ica*-dependent biofilm formation appears more robust, *S. aureus* biofilm development largely relies on protein production rather than the biosynthesis of the *ica* locus products [36]. Consequently, the complexity of biofilm structures confers enhanced resistance to multiple antibiotics, rendering biofilm-forming strains significantly more difficult to treat compared with non-biofilm producers [37,38].

### 1.3. Genes Associated with Antimicrobial Resistance and Mobile Genetic Elements

The dissemination of resistance determinants necessitates consideration of the role of mobile genetic elements (MGEs) and the mechanisms that govern their mobility. MGEs are DNA sequences that facilitate both intracellular mobility (e.g., from a chromosome to a plasmid or between plasmids) and intercellular DNA mobility [39,40,41]. In staphylococci, these elements primarily include transposons, insertion sequences, and integrons [42]. Insertion sequences (ISs) and transposons (Tns) are discrete DNA segments capable of relocating (often carrying associated resistance genes) within the same or between different DNA molecules within a single cell or between different cells [43,44]. In contrast, integrons (Ins) utilize site-specific recombination to transfer resistance genes between predefined loci [43,45]. Because MGEs are often present in multiple copies at various genomic sites, they can also promote homologous recombination, wherein identical or closely related DNA sequences are exchanged [46]. Intercellular genetic exchange occurs via mechanisms such as conjugation/mobilization (mediated by plasmids or integrative conjugative elements), transduction (mediated by bacteriophages), and transformation (the uptake of extracellular DNA) [42,47].

Interactions among different types of MGEs significantly drive the rapid evolution of multidrug-resistant pathogens. In this regard, a variety of genes and their associated MGEs have been identified in *S. aureus* [45,46,47]. For instance, the Tn558 transposon harbors several antimicrobial resistance genes, including the transferable multidrug resistance gene *cfr*, the chloramphenicol resistance gene *fexA*, the aminoglycoside resistance genes *aacA-aphD* and *aadD*, and the bleomycin resistance gene *ble* [44,48]. The Tn558 transposon has been identified in the livestock-associated MRSA (LA-MRSA) strain ST398 [48]. Beyond Tn558 transposon, the *cfr* gene has also been identified in a plasmid, pSCFS7, in an ST8-MRSA/USA300 isolate [49]. In this strain, the *fexA* gene was likewise observed, with both genes linked to plasmids such as pSCFS3 and pSCFS6 [49,50]. Notably, Shore et al. also detected SCC*mec* IVa in the ST8-MRSA/USA300 strain [49] which, in addition to harboring the *mec*(A-D) gene, contains other genes associated with resistance and virulence [4,51]. This suggests that SCC*mec* may play a more significant role in the pathogenic evolution of *Staphylococcus* than previously appreciated.

Another important gene in staphylococci is the fusidic acid (FA) resistance gene and its homologs [45,52]. In *S. aureus*, FA is a key antibiotic used to treat infections caused by MRSA, VISA, or VRSA, with resistance to this antibiotic still being relatively rare [53,54]. FA resistance in *S. aureus* arises either through chromosomal mutations in *fusA* (encoding elongation factor G, a therapeutic target of FA) or through the acquisition of horizontally transferred elements, such as the FA resistance gene 1 (*far1* or *fusB*) and its homologs (*fusC* and *fusD*) [55]. In instances of acquired resistance, the plasmid pUB101, which carries the *far1* gene, has been identified in *S. aureus* strains isolated from hospital environments [56].

### 1.4. Whole Genome Sequencing (WGS) Technology and the Formation of Big Data

Whole genome sequencing (WGS) technology has emerged as a powerful tool for the surveillance of pathogens from human, animal, and environmental sources [57]. This approach generates vast amounts of data from a single assay, enabling the simultaneous execution of multiple in silico analyses. In contrast to phenotypic sensitivity testing, which is often time-consuming, labor-intensive, and highly dependent on standardized laboratory protocols, WGS data can be processed and analyzed with greater speed and efficiency [58]. Numerous studies have highlighted the effectiveness of WGS in thoroughly identifying and characterizing infectious agents during outbreaks, as well as in identifying genotypes associated with antimicrobial resistance, virulence, and biofilm formation in pathogens isolated from diverse environments [59,60].

Recent years have witnessed a substantial increase in sequencing data, driven largely by the decreasing cost of sequencing technology, particularly the adoption of low-cost Illumina platforms. These platforms generate shotgun sequencing data from short reads, which cannot be directly assembled into a single DNA contig but can be aligned to a reference genome from the same lineage [61]. While this technological advancement has significantly enabled our understanding of genetic variation, gene expression, and microbiome dynamics, it also introduces practical and conceptual challenges [62]. These include issues related to data manipulation, storage, and statistical analysis, as well as the inherent complexity of analyzing heterogeneous cell populations [63].

The growth of large-scale databases (Big Data), particularly microbial genomic databases, can be subject to selection biases due to the preference for sequencing isolates of greater clinical or industrial relevance [64]. Nevertheless, the extraordinary volume of available data offers an unparalleled resource for studying trends in the evolution and epidemiology of staphylococcal isolates, despite the potential for selective bias in the choice of isolates [59,61].

### 1.5. The One Health Concept and the Evolution of S. aureus

Within the framework of the One Health concept (Figure 2), which emphasizes the interconnectedness of human, animal, and environmental health, *Staphylococcus aureus* is a pathogen of significant concern [65]. The species is characterized by several alarming features, including antimicrobial resistance, intricate virulence mechanisms, biofilm formation, and widespread distribution. Since its initial identification by Alexander Ogston in 1888, *S. aureus* has evolved from a penicillin-sensitive microorganism to one that exhibits resistance to multiple classes of antimicrobials, such as macrolides, fluoroquinolones, aminoglycosides, tetracyclines, and lincosamides [66,67].

This alarming scenario can be attributed to a combination of factors, including bacterial fitness, genetic mutations, and the acquisition of mobile genetic elements (mobilomes) [65,68]. Additionally, the overuse and misuse of antimicrobials in healthcare settings, communities, and the agricultural sector have contributed to the diminishing efficacy of these drugs, fostering the emergence of resistant strains [68]. The global rise in antimicrobial resistance (AMR) poses a profound public health threat, as AMR bacteria complicate the management of infections and disease treatment [65]. Furthermore, resistant bacteria can enter the food chain via consumption, thereby increasing the risk of AMR in foodborne pathogens, which can, in turn, return to the community, exacerbating the cycle of infection [69].

In this cyclical process, food-producing animals and food products may become contaminated with AMR microorganisms through human handlers [70,71,72]. This pathway perpetuates the spread of AMR microorganisms, resulting in morbidity in both animals and humans, while environmental contamination further amplifies the risk by sustaining the cycle of transmission [73]. In response to the escalating threat of resistance, the One Health approach has been adopted globally [70,73,74]. Numerous countries have implemented national action plans aligned with the One Health concept to combat AMR, by the guidelines established by the quadripartite collaboration of the Food and Agriculture Organization (FAO), the World Organization for Animal Health (WOAH), the World Health Organization (WHO), and the United Nations Environment Programme (UNEP) [73]. In 2015, the WHO, in partnership with the FAO and WOAH, published the Global Action Plan (GAP) on Antimicrobial Resistance, which set a global strategy to tackle AMR and provided a foundation for the creation of national action plans in individual countries [65,73].

Despite the efforts of international organizations, substantial gaps persist, particularly in developing countries [65,69,73,74]. These gaps include limited surveillance capacity, particularly in genomic monitoring; inadequate measures to optimize antimicrobial use and strengthen infection prevention and control programs; and, most critically, the increased use of antimicrobial use in livestock [65,68,73]. Considering these challenges, the current study seeks to evaluate the global dissemination of genes linked to biofilm production and antibiotic resistance in strains isolated from diverse sources worldwide, using assembled *S. aureus* sequences available at the NCBI. Additionally, this study will explore a possible correlation between these genes and their global distribution.

## 2. Results

### 2.1. Analysis of S. aureus Genomes: Geographic Distribution, Clonal Complexes, and SCCmec Isotypes

A total of 94,600 *S. aureus* genomes were identified and downloaded from the NCBI Pathogen Detection database. Of these, 44,069 presented metadata and were selected in the screening stage (Table 1). Of these, 42,188 were human clinical samples and 1881 were environmental samples (Table 1). Regarding the first category, we found that the largest number of genomes analyzed was observed in Europe (38%, 15,957 genomes) and North America (38%, 15,832 genomes), followed by Asia (11%, 4674 genomes), Oceania (8.3%, 3496 genomes), South America (3.3%, 1372 genomes), and Africa (2.0%, 857 genomes). Concerning the environmental origin category, Asia (47%, 889 genomes) presented the largest number of samples, followed by North America (25%, 477 genomes), Europe (22%, 408 genomes), Africa (3.7%, 70 genomes), South America (1.2%, 22 genomes), and Oceania (0.8%, 15 genomes).

The most prevalent clonal complexes identified in human samples were ST8 (18%) (commonly referred to as USA300), ST5 (15%), ST22 (7.4%), and ST30 (4.5%). In contrast, among environmental isolates, ST398 was the most dominant (17%), followed by ST59 (9.3%), ST8 (8.6%), and ST5 (7.8%) (Table 1). The SCC*mec*-associated genes *mec*A and *mec*C2 were detected at an identical frequency of 58%. Notably, a second *mec*C gene was identified with a frequency of 35%. This additional *mec*C gene is classified among the resistance genes documented in the metadata of the *S. aureus* sequences (Table 2). Regarding SCC*mec* isotypes, only types IV, IVa, and II exhibited substantial frequencies: 32%, 22%, and 12%, respectively (Table 2), specifically, 32%, 22%, and 13% in clinical samples and 21%, 17%, and 5.5% in environmental samples (Appendix A).

### 2.2. Biofilm Genes

A total of 140 genes associated with biofilm production were analyzed across all genomes. Among these, 20 genes exhibited a statistical frequency exceeding 50% (Table 2). The highest frequencies were observed for *ebpS* (100%), *clfA1* (100%), *clfB1* (99%), and *eap/map* (99%). Additionally, all genes within the *ica* locus and the *isd* block demonstrated a frequency of 100%. Other notable genes, including *fnbA*2, *sarA*, and *fnbB*1, were detected at frequencies of 98%, 95%, and 81%, respectively.

### 2.3. Correlation Between Resistance Genes and Biofilm-Associated Genes

In the metadata, 249 resistance-related genes were identified. To elucidate the results, homologous genes were grouped: *drf*, *fexA-B*, *fusA*, and *aph*. The genes with the highest frequencies across all sequences were *drf* (50%), *fexA-B* (45%), *fusA* (29%), *aph* (28%), *ant(1″)-Ia* (27%), *aac(6′)-Ie/aph(1″)-Ia* (22%), *tet, abc-f,* and *cfr* (15%) (Table 2). In addition to the previously mentioned *mec*A, *mec*C1, *mec*C, and SCC*mec* type IIa, IVa, and IV genes, we conducted a correlation analysis between resistance genes and biofilm-associated genes (Figure 3).

The group of genes related to the PIA polysaccharide—*ica*C1, *ica*R2, *ica*B, *ica*A2, and icaD1—demonstrated a high level of correlation among themselves. Remarkably, the *sasG* gene (*S. aureus* surface protein G) was the only biofilm-related gene to show a connection with resistance genes. The *crf, mec*A*, mec*C1, and SCC*mec* type IIa genes were linked to *sasG*, as illustrated in the dendrogram and correlation values (Figure 3). Among the resistance genes, *mecA* and *mecC2* showed a strong correlation, as did *abc-f* and *ant(2″)-Ia* (Figure 3). Another interesting finding is that the *fexA*-*B*, *fusA*, *aac(6′)-Ie/aph(1″)-Ia*, and *tet* genes also exhibited a high correlation, forming a distinct cluster separate from the other genes (Figure 3).

### 2.4. Gene Clustering and Correlations in Human Clinical Sequences

When analyzing only the human clinical sequences, it is evident that the clustering of the *fexA-B*, *fusA*, *aac(6′)-Ie/aph(1″)-Ia*, and *tet* genes persists, showing a strong correlation (Figure 4). The *dfr*, *aph*, and *mecC* genes also demonstrate a correlation, and a group of *cls* genes (associated with daptomycin resistance) appears in this analysis. Another significant finding is the relevance of SCC*mec* type II, which correlates with SCC*mec* type IIa, *sas*G, and *cfr*. The remaining correlations align with the overall analysis (Figure 4).

### 2.5. Gene Clustering and Correlations in Environmental Sequences

When the sequences from environmental samples were evaluated, a greater variety of resistance-linked genes was observed. Genes *vga(E)*, *erm*, *gyrB_D416N*, *parC_E84V*, *msrF*, *mprF_L141S*, and *rpo*, which did not show significance in previous analyses, were now identified (Figure 5). Additionally, these genes were grouped with a high correlation alongside *fusA-B*, *fexA-B*, *aac(6′)-Ie/aph(1″)-Ia*, and *tet*, which were also linked to previous analyses. Another notable observation is that the *sasG* gene, which previously showed a correlation with resistance genes in global sequences, is now linked to other biofilm production genes (Figure 5). Changes in the SCC*mec* correlations were also observed here. SCC*mec* types V and VII, which had not shown significance in the global and human clinical analyses, now appeared correlated. These SCC*mec* types (V and VII) exhibited a correlation between *dfr* and *mec*C genes (Figure 5). The remaining correlations resemble those from the human clinical analyses (Figure 3 and Figure 4).

### 2.6. Global Distribution of MLST in Human Clinical and Environmental Sequences

Figure 6 presents the distribution of sequence types (STs) across clinical and environmental isolates on different continents, depicted as proportional bar plots. Each panel represents a continent, facilitating a comparative analysis of isolate distribution worldwide. Specific STs are color-coded, while gray denotes sequences classified as “NA” (not assigned).

In Africa, ST5 predominates in human clinical isolates, whereas ST30 is the most frequent among environmental isolates. A considerable proportion of ST8 was also detected in environmental sequences on this continent. Asia exhibits a pronounced dominance of ST8 and ST239 in both clinical and environmental isolates, with environmental sequences displaying a broader range of STs. Europe shows a significant presence of ST239 and ST398 in clinical isolates, while environmental isolates contain a higher proportion of NA sequences.

In North America, ST8 and ST15 are the most prevalent among clinical isolates, whereas environmental isolates exhibit greater diversity, with notable contributions from ST45 and ST239. Oceania displays a more balanced distribution across both isolate types, with ST8, ST15, and ST398 being the most prominent. In South America, ST5 and ST8 are strongly represented in clinical isolates, whereas environmental isolates show a higher proportion of ST239 and NA sequences (Figure 6).

These findings highlight regional variations in ST distribution and underscore differences between clinical and environmental isolates across continents. The observed diversity reflects the adaptability of *S. aureus* across distinct ecological niches and geographical regions.

### 2.7. Global Distribution of SCCmec Types Among MLST Profiles and Continents

The heatmap in Figure 7 illustrates the distribution of SCC*mec* types across various multilocus sequence types (MLSTs) from different continents, covering both human clinical and environmental isolates. The intensity of the shading represents the relative frequency of SCCmec elements within each MLST–continent combination, with a gradual transition from white to red indicating higher frequencies. 

Notably, SCC*mec* types IV and V are predominantly associated with MLSTs from North America and Europe, aligning with their established prevalence in these regions. In contrast, types VII and IX are more frequently observed in isolates from Oceania and Asia.

ST398, commonly linked to livestock-associated MRSA, exhibits a global distribution, particularly in Europe and Asia, with a diverse range of SCC*mec* types. Additionally, MLST 8 shows a strong correlation with SCC*mec* type IV and its subtype IVa in sequences from North America. In Europe, ST239 is primarily associated with SCC*mec* type IV and its subtype IVh, whereas in Asia, ST239 correlates with SCC*mec* types II, V, VII, and IIIa.

These findings emphasize regional variations in SCC*mec* distribution and highlight the genetic diversity of MLSTs associated with these elements on a global scale.

## 3. Discussion

### 3.1. Distribution of Sequences in the World

The metadata analysis of *S. aureus* sequences highlights a pronounced disparity in the availability of genomic data across global regions. Among the 44,069 sequences examined, 76% originate from Europe and North America, while the remaining 24% collectively represent South America, Africa, Asia, and Oceania. This imbalance underscores the persistent inequities in the adoption and dissemination of sequencing technologies, as well as the application of bioinformatics tools for genome analysis and assembly. To mitigate potential biases arising from recognized gaps in global microbial surveillance, South America was analyzed separately from the broader Americas. For instance, Brazil—a leading producer of animal protein and a nation actively addressing the rise of antimicrobial resistance (AMR)—contributed only approximately 380 sequences, encompassing both environmental and human samples. This limited representation is concerning, particularly given the emergence of new AMR hotspots in critical animal production regions [65].

The dissemination of the USA300-ST8 clone of MRSA in Latin America underscores its significance as an emerging pathogen in both community-acquired and healthcare-associated infections. A multicenter study conducted between 2006 and 2008 across four Andean countries revealed a high prevalence of MRSA, with the highest rates observed in Peru and Colombia. While the Chilean clone (ST5-SCC*mec* I) dominated healthcare-associated infections, the USA300-ST8 clone emerged as the primary cause of community-acquired infections, particularly in Ecuador and Colombia. The study highlighted the genetic adaptability of the USA300-ST8 clone, including SCC*mec* IV variants and tetracycline resistance, and emphasized the necessity of molecular surveillance to curb its spread in the region [75].

The epidemiological and genetic evolution of infections caused by MRSA and MSSA in Argentina between 2009 and 2015 was also examined. During this period, the overall incidence of *S. aureus* infections increased significantly, driven primarily by community-acquired MSSA infections, while MRSA rates remained stable. Emerging clones, such as CA-MRSA (community-associated MRSA) ST30-IV and CA-MRSA ST5-IV, gained prominence in both community and healthcare settings, partially displacing traditional healthcare-associated clones like HA-MRSA (hospital-associated methicillin-resistant *S. aureus*) ST5-SCC*mec* I. The study also documented a rise in antimicrobial resistance among MSSA strains, particularly to erythromycin, and underscored the need for continuous molecular surveillance to elucidate the dynamics of these pathogens and enhance control and management strategies, especially considering increasing infection rates in both community and healthcare settings [76].

Despite these critical insights, genomic studies of pathogens in Latin America remain scarce. A significant gap exists in the systematic application of genomic surveillance to monitor the spread and evolution of resistant strains across the region, impeding the development of effective strategies to address these emerging threats.

Another notable observation is the disproportionate sequencing of human clinical samples compared with environmental, food, or animal clinical samples. Many studies justifiably focus on clinical isolates, particularly in the context of hospital outbreak investigations. In such cases, the primary objective is to characterize pathogenic strains, track the spread of antimicrobial resistance, and implement infection control measures.

Given the urgent need to contain outbreaks and protect public health, clinical samples are prioritized over environmental or non-clinical isolates. Additionally, healthcare-associated *S. aureus* strains often exhibit higher selective pressures, leading to the emergence of multidrug-resistant clones that pose significant therapeutic challenges.

While this clinical focus is necessary, it may lead to an underrepresentation of environmental reservoirs that contribute to pathogen persistence and transmission. Previous studies have demonstrated that identical *S. aureus* isolates can be recovered from patients, healthcare workers, and hospital surfaces, highlighting the importance of a holistic surveillance approach that integrates both clinical and environmental sources [77,78,79,80].

Of the total sequences analyzed, 42,188 were derived from human clinical samples, while only 1881 originated from environmental sources. This disparity complicates efforts to compare these groups and draw reliable conclusions. Furthermore, out of 94,600 sequences initially identified, 50,531 were excluded due to incomplete metadata, rendering approximately 53% of the data unusable owing to insufficient epidemiological information.

The global incidence of *S. aureus* in animals, food, and environmental sources has been extensively documented, raising significant public health concerns. This pathogen is prevalent in various animal species, particularly cattle, pigs, and poultry, and can colonize tissues and secretions such as milk and meat. Transmission to humans can occur through the consumption of contaminated food or direct contact with infected animals. Additionally, *S. aureus* has been detected in healthcare, agricultural, and domestic environments, where contamination often results from inadequate hygiene and food handling practices [81,82,83,84].

Antimicrobial resistance in *S. aureus* strains, particularly MRSA, poses a significant challenge, as it can compromise treatment efficacy and exacerbate infection severity. Although epidemiological and resistance studies of *S. aureus* exist across various contexts, the scarcity of genomic research limits a comprehensive understanding of the evolution and dissemination of resistant strains. There is an urgent need for expanded genomic studies to more accurately trace the origins, resistance *mec*hanisms, and transmission dynamics of *S. aureus*, particularly in veterinary and food-related contexts, to inform the development and implementation of effective global control measures [85,86,87,88].

The absence of standardized protocols for sharing sequencing data on public platforms further complicates efforts to monitor and track clonal complexes of interest. For example, without critical metadata such as isolate origin, phenotypic resistance profiles, host information, or environmental context, it is impossible to make accurate inferences or associations following the isolation and sequencing of pathogens from infectious outbreaks.

In an increasingly globalized world—where food trade, human mobility, and environmental factors such as pollution significantly influence the spread of microorganisms—it is imperative that microbial surveillance be conducted on a global scale. The widespread adoption of genomic tools, coupled with the establishment of comprehensive genomic databases that include detailed epidemiological data, is essential for tracking pathogenic microorganisms and guiding public health interventions.

### 3.2. Association of Genes Linked to Biofilm Production and Resistance Genes

Concerning the associations between genes, the *ica*ADBC operon, responsible for encoding the production of PNAG or PIA, demonstrated a strong correlation among its constituent genes, indicating the presence of the complete operon (Figure 3). This was observed in both environmental and human clinical sequences. Approximately 44,000 sequences contained the *ica* operon, underscoring its widespread dissemination among *S. aureus* strains, a finding consistent with the existing literature [37].

The *ica*ADBC operon plays a pivotal role in biofilm formation by *S. aureus*, particularly in MRSA. This operon facilitates the synthesis of PIA, a polysaccharide critical for bacterial adhesion and biofilm development. In MRSA strains, the expression of the *ica*ADBC operon is frequently associated with enhanced biofilm-forming capacity, contributing to bacterial persistence and resistance to antibiotic therapies. The regulation of this operon is complex and can be influenced by both environmental and genetic factors, including the presence of the *mec*A gene, which confers methicillin resistance [37,38,89].

Given the extensive dissemination of the *ica*ADBC operon observed in this study, it is plausible to hypothesize that the exchange of mobilomes between environments may facilitate the spread of this gene complex. However, the intricate architecture of *S. aureus* biofilms means that the mere presence of the *ica*ADBC operon does not guarantee biofilm formation. Nonetheless, it suggests gene flow between distinct clonal complexes. A deeper understanding of the regulatory *mec*hanisms governing the *ica*ADBC operon is essential for developing effective therapeutic strategies against *S. aureus* infections.

The *isd* gene cluster was predominantly identified in human clinical strains, aligning with its role in iron acquisition by *S. aureus*. The *isd* system enables the bacterium to bind to human hemoproteins, extract heme molecules, and transport them across the cell wall and plasma membrane for accumulation in the cytoplasm [90]. However, studies have demonstrated that the inactivation of Isd surface proteins does not necessarily impair iron uptake in *S. aureus* [91]. Recent research further suggests that Isd proteins are more involved in the initial colonization of host tissues than in heme-derived iron acquisition [91,92,93].

The proteins ClfB, IsdA, SdrC, and SdrD were also analyzed. Using the *S. aureus* Newman strain and mutants deficient in one or more of these proteins, researchers demonstrated that the absence of all four proteins significantly reduced bacterial adhesion. Complementation experiments with plasmids encoding these proteins confirmed their critical role in adhesion [94]. It is hypothesized that Isd proteins may facilitate bacterial attachment and invasion via platelet integrins, with adhesion and invasion occurring independently of hemoglobin binding.

While MRSA isolates exhibit resistance to most β-lactam antibiotics, there is ongoing debate regarding whether they possess unique virulence factors that could exacerbate infections [13,95]. Notably, the presence of such factors appears to be linked to the origin of the isolates. For instance, the Bap protein has been identified in isolates from bovine mastitis but is absent in human isolates [31,96,97]. In this study, the *bap* gene, responsible for encoding the Bap protein, was not detected in any of the analyzed sequences, likely due to the high proportion of sequences derived from human isolates.

Interestingly, some studies have shown that MSSA isolates primarily form biofilms via a PIA-dependent mechanism, whereas MRSA isolates rely on a PIA-independent pathway involving LPXTG motif proteins and extracellular DNA (eDNA) [38,41,98]. These LPXTG proteins, such as SasG, are anchored to the cell wall by the enzyme sortase A [98] and are essential for biofilm formation in MRSA. Genetic analyses have revealed a distinct repertoire of LPXTG motif proteins in MRSA, with several being identified as critical for biofilm formation, depending on the strain [32,36,93,99].

Additionally, while some studies have associated biofilm formation with specific SCC*mec* types, others argue that the origin of the isolates is more significant than the presence of SCC*mec* elements. It is evident that, in *S. aureus*, virulence and resistance are closely linked to clonal complexes, posing a substantial threat to both human and animal health [1,41,100].

It is imperative to acknowledge that antimicrobial resistance (AMR) represents a growing global public health crisis, with dire projections for the coming decades. Research indicates that, without effective interventions, AMR could be responsible for up to 10 million deaths annually by 2050, surpassing fatalities from diseases such as cancer [73]. Furthermore, AMR is predicted to have a catastrophic economic impact, with potential global losses of up to USD 100 trillion by 2050. Beyond its economic burden, AMR threatens advancements in modern medicine and endangers millions of lives worldwide [73].

### 3.3. Correlation Between Antimicrobial Resistance Genes and Biofilm-Related Genes

A notable finding is the association of the *sasG* gene, among biofilm-related genes, with both resistance genes and the SCC*mec* type IIa element in environmental and human clinical sequences (Figure 3). Despite its relatively low prevalence among global sequences—approximately 5635 (13%)—its correlation with resistance genes remained evident (Table 2 and Figure 3).

The *sasG* gene plays a critical role in the initial attachment to surfaces and host tissues during infection [98]. It has also been implicated in the biofilm-forming capacity of *S. aureus*, with its dissemination across various clonal complexes facilitated by genomic islands (GIs) [40,98].

Genomic islands are defined as large genomic regions within bacterial genomes, likely acquired through horizontal gene transfer [101]. SCC*mec* elements align with this definition [102], supporting the hypothesis that *sasG* may be carried by SCC*mec*, which can itself be considered a genomic island. This hypothesis is consistent with the observed correlation (Figure 3). Furthermore, prior studies have identified plasmids such as pLC1, SCC*mec* type IV, and insertion sequences like IS431 (within SCC*mec*) as carriers of the *sasG* gene [35,98,102].

Another significant observation is the role of the SasG protein as a key virulence factor, associated with bacteremia and enhanced biofilm formation, thereby promoting bacterial survival in both host and hospital environments. It is also linked to clonal complexes of clinical relevance in human infections [98]. In this study, the *sasG* gene exhibited a strong correlation with the *cfr* gene, a critical association given that *cfr* confers a multidrug-resistant phenotype. The *cfr* gene encodes a methyltransferase that mediates post-transcriptional methylation of the 23S rRNA at position A2503, impairing the binding of at least four antimicrobial classes: phenicols, lincosamides, pleuromutilins, and streptogramin A [103]. This gene can be carried by transposons (Tn558) and plasmids (pSCFS7), suggesting potential co-circulation of these resistance genes between lineages or specific clonal complexes. However, further research is necessary to fully elucidate the interplay between biofilm production genes, antimicrobial resistance genes, and specific mobilomes.

### 3.4. Acquisition of Antimicrobial Resistance in Clinical Strains

In this study, two distinct patterns emerged when analyzing antimicrobial resistance genes in human clinical sequences. First, compared with environmental sequences, human clinical sequences displayed reduced diversity in resistance-associated genes. Second, certain genes exhibited significant correlations, suggesting their co-occurrence within the analyzed sequences. As illustrated in the heatmap (Figure 4), the genes *fexA-B*, *fusA*, *aac(6′)-Ie/aph(2″)-Ia*, and *tet* demonstrated strong correlations, indicating their frequent co-presence. These genes confer resistance to multiple antimicrobial classes, including phenicols, macrolides, and tetracyclines, among others [73,74].

Additionally, other resistance genes, including *cls* (conferring resistance to daptomycin), *dfr* (resistance to trimethoprim), *aph* (resistance to aminoglycosides), and *cfr* (conferring multidrug resistance to phenicols, lincosamides, pleuromutilins, and streptogramins type A) also showed significant correlations [46,48,104,105].

This particular *cls* gene exhibited a notable correlation with SCC*mec* type IV and IVa chromosomal cassettes. SCC*mec* type IV, especially subtype IVa, is known for its role in transporting plasmids, genomic islands, and insertion sequences, functioning as a mobile genetic element. This type has been detected across diverse genetic backgrounds, suggesting greater mobility compared with larger SCC*mec* types [1,106]. Subtype IVa is distinguished by the presence of a pUB110 plasmid, which enhances its mobility [1,107].

The pEF332-2 plasmid has also been implicated in the carriage of the *cls* gene, indicating that this association may be mediated by mobile genetic elements, potentially SCC*mec* type IV or IVa [1,102]. A similar pattern was observed for the correlation between the *cfr* gene and SCC*mec* types II and IIa, which also showed significant associations in this study. SCC*mec* type II has been previously linked to the Tn554 transposon, which carries the *ermA* (erythromycin resistance) and *spc* (spectinomycin resistance) genes [1,43,102].

The rise in antimicrobial resistance among strains isolated from both nosocomial and community settings is driven by factors such as self-medication, incorrect dosages, prolonged antibiotic use, lack of standardized protocols for healthcare professionals, and the presence of antibiotics as environmental pollutants [13,74,108]. This resistance poses a significant challenge to controlling HA-MRSA infections, including surgical site infections, pneumonia, and bloodstream infections. The increasing prevalence of CA-MRSA infections further highlights the growing overlap between community and hospital strains [48,108].

In response to the reduced susceptibility of MRSA strains to vancomycin, daptomycin, a cyclic lipopeptide effective against MRSA, has been proposed as an alternative [109,110]. Consequently, the observed association between the *cls* gene, which confers daptomycin resistance, and SCC*mec* type IV, which has rapidly disseminated among HA-MRSA and CA-MRSA strains, is particularly concerning. Linezolid, the first oxazolidinone used clinically, has demonstrated potent antimicrobial activity against Gram-positive pathogens, including MRSA [103]. However, the *cfr* gene, which confers resistance to linezolid and promotes multidrug resistance to other classes of antibiotics (e.g., phenicols and lincosamides), was found to be associated with SCC*mec* types II and IIa in this study. These SCC*mec* types have been linked to pandemic *S. aureus* strains of nosocomial origin, suggesting that the *cfr* gene is carried by these elements and contributes to multidrug resistance.

The observed correlations between specific resistance genes and mobile genetic elements, such as SCC*mec* types IV, IVa, II, and IIa, underscore the critical role of these elements in disseminating resistance traits across bacterial populations. The significant associations of the *cls* and *cfr* genes with distinct SCC*mec* elements highlight the complexity of resistance gene mobility and its implications for public health. Further research into the dynamics of these genetic elements and their interactions with resistance determinants will be essential for developing more effective strategies to combat antimicrobial resistance.

### 3.5. Acquisition of Antimicrobial Resistance in Environmental Strains

A notable finding of this study is the comparison of resistance gene frequencies between human clinical and environmental samples. Environmental samples exhibited a greater diversity of resistance genes (Figure 5, Appendix A). Genes such as *vga(E)*, *erm*, *gyrB_D416N*, *parC_E84V*, *msrF*, *mprF_L141S*, and *rpo* were observed at higher frequencies in environmental sequences but did not show statistically significant prevalence in global or human clinical sequences (Figure 3, Figure 4 and Figure 5 and Appendix A). These genes encode resistance mechanisms to various antimicrobial classes, including aminoglycosides, phenicols, triterpenoids, and cyclic lipopeptides, among others [45,48,111,112].

In this context, it is important to emphasize that scientific evidence increasingly indicates a rise in resistance levels among environmental isolates due to the uncontrolled use of antimicrobials and environmental pollution [65,72,113,114]. Antibiotic use in animal production for human consumption was estimated at 63,151 (±1560) tons in 2010, with projections indicating a 67% increase to 105,596 (±3605) tons by 2030 [65,114,115]. Two-thirds of this global increase (66%) in antimicrobial consumption is attributed to the rising number of animals raised for food production. By 2030, antimicrobial consumption in Asia is projected to reach 51,851 tons, accounting for 82% of the global antimicrobial use in food animals [59,114,115].

In 2010, the five countries with the highest shares of global antimicrobial consumption in food animal production were China (23%), the United States (13%), Brazil (9%), India (3%), and Germany (3%). By 2030, the projected ranking is China (30%), the United States (10%), Brazil (8%), India (4%), and Mexico (2%) [68,114]. Alongside the alarming data on antibiotic use in food production, environmental pollution has also escalated significantly due to anthropogenic activities [116]. Solid and liquid waste from urban areas poses a global challenge in pollution management. High loads of antibiotics, chemicals, and micropollutants released into the environment through anthropogenic waste alter the biodiversity of microbial ecosystems [116]. The combination of these practices may explain the observed diversity of antibiotic resistance genes in environmental sequences found in the present study.

Furthermore, one hypothesis to consider is the role of the SCC*mec* element in harboring additional genes beyond *mec*(A-E) within its J regions [1,116]. This underscores SCC*mec* as a highly mobile chromosomal cassette frequently detected in circulating *S. aureus* strains, potentially facilitating the acquisition of supplementary resistance and virulence determinants. Such a mechanism drives the evolution of clonal complexes that already pose significant public health concerns [1,37,116]. Although the number of environmental genomes analyzed was comparatively lower than that of human clinical isolates, the findings herein reveal a greater diversity of resistance genes within environmental samples. This observation aligns with the existing literature, which similarly underscores the increasing heterogeneity of resistance determinants in non-clinical settings.

### 3.6. Prevalent Sequence Types (STs) in Human Clinical and Environmental Samples

In the investigation of clonal complexes, this study identified ST8 and ST5 as two of the most prevalent lineages in sequences derived from human clinical samples. ST8, commonly referred to as the USA300 strain, represents an amalgamation of two distinct genetic backgrounds [67]. The high proportion of North American sequences in the NCBI database likely accounts for its predominance, as the USA300/ST8 clone is particularly widespread in the United States and Canada, especially within human populations [67].

The emergence of the ST8 clonal complex warrants particular attention. This virulent clone was first identified in the United States around the turn of the millennium. Initially detected predominantly in specific at-risk populations, it rapidly disseminated throughout the broader community, eventually becoming the dominant CA-MRSA strain in North America [49]. In 2003, it was designated “USA300” based on its distinctive pulsed-field gel electrophoresis (PFGE) pattern [67].

The classical USA300 clone is characterized by a remarkable assembly of genetic attributes, including its association with ST8 and spa type t008, the absence of functional capsular polysaccharides due to point mutations in the *cap5D* and *cap5E* genes, and the presence of SCC*mec* IVa, PVL-encoding genes (*lukF*-PV *and lukS*-PV); and the arginine catabolic mobile element (ACME) [11,117]. Initially, ST8-MRSA-IVa isolates harbored only the *mec*A and erythromycin resistance genes. However, over time, they have disseminated globally, acquiring additional resistance determinants such as *erm*(A), *erm*(C), *tet*(M), *tet*(K), and *mupA*, which confer resistance to macrolides-lincosamides-streptogramin B, tetracyclines, and mupirocin, respectively. Most of these resistance determinants are located on plasmids [49].

Regarding ST5, a human-associated clonal lineage initially identified in Poland, this strain underwent a significant evolutionary transition, enabling it to infect poultry raised for human consumption [118]. This evolutionary adaptation enhanced ST5′s capacity to evade host immune responses, particularly those employed by chicken heterophils [88,118]. Consequently, it emerged as a prominent animal pathogen, exhibiting rapid intercontinental dissemination [88,119]. This adaptability likely explains why ST5 ranks as the second most frequently observed sequence type (ST) in human clinical isolates and the fourth most frequent in environmental samples (Table 1, Figure 6). The ST5 clone of *S. aureus* is widely recognized for its extensive global dissemination and strong association with hospital-acquired infections [119,120]. Research indicates that ST5 is linked to the New York/Japan clone (ST5-II), distinguished by its methicillin resistance and broad geographical distribution [54]. Furthermore, ST5 is frequently associated with the production of staphylococcal enterotoxins, which play a significant role in foodborne illnesses. Notably, *S. aureus* ST5 is among the most frequently reported clones in Asia [121]. Emerging evidence also suggests that this lineage has extended beyond hospital environments into community settings, contributing to the rise of CA-MRSA [119].

With respect to ST1, this clonal complex is predominantly associated with humans, particularly with Panton–Valentine leukocidin (PVL)-positive CA-MRSA [122,123]. In the United States, this lineage is designated USA400. It has been detected in international travelers, including a Dutch patient (2005–2006), Italian patients (2007), and cases in Denmark [124]. Currently, ST1 is among the most commonly isolated strains in the Italian swine industry and has also been identified as a causative agent of bovine mastitis in dairy cattle and as a contaminant in bulk goat milk [125]. This represents yet another example of a human-associated clonal complex successfully adapting to animal hosts and causing serious infections, reinforcing the significance of the One Health approach, which underscores the necessity of enhanced surveillance of zoonotic pathogens.

In the present study, ST1 ranked eighth in absolute frequency among all sequences (Table 1) and was detected in environmental samples from the Americas (North and South), Asia, and Oceania. Notably, in Oceania, it was also identified in clinical isolates (Figure 6). The USA400 clone of *S. aureus*, synonymous with ST1, is an MRSA lineage primarily linked to community-acquired infections. This clone is defined by the presence of genes encoding PVL, a toxin that promotes leukocyte destruction and is associated with severe skin infections and necrotizing pneumonia. Additionally, USA400 harbors the SCC*mec* type IV element, conferring methicillin resistance. A study by demonstrated that while USA400 was one of the earliest community MRSA strains identified in the United States, it has largely been supplanted in prevalence by the USA300 clone [103]. However, genes regulated by the *agr* (accessory gene regulator) system—such as *sae*RS (*S. aureus* exoprotein expression regulatory system), *hla* (alpha-hemolysin gene), *luk*S-PV (Panton–Valentine leukocidin), and two PSMs (phenol-soluble modulins) essential for infection—are expressed at higher levels in USA400 compared with the USA300 complex. This observation underscores the high virulence potential of USA400 in human infections. Furthermore, the expression of these virulence-associated genes varies significantly depending on the growth phase or stage of infection, suggesting potential differences in pathogenicity and an increased virulence profile in USA400.

ST22, the third most frequently identified ST among human clinical isolates in this study, rose to prominence in Europe during the 1990s [126]. Currently, it represents the most prevalent strain in hospital settings in England and continues to be detected in healthcare environments across Asia [126]. Notably, in 2003, the first ST22-IV clone was identified in Asia, demonstrating high susceptibility to several non-β-lactam antibiotics in comparison with more concerning clonal complexes, such as ST239 [127]. By 2013, a substantial proportion of ST22 MRSA isolates exhibited high levels of resistance to mupirocin, an antibiotic introduced in 1996 specifically for MRSA decolonization [125,127]. Furthermore, in 2004, ST22-IV emerged as the predominant strain in Singapore, and during the same period, this clone was first identified in a pig from Indonesia [127].

By 2010, the highly transmissible HA-MRSA clone ST22-IV had been reported in Malaysian hospitals, exhibiting substantial resistance to ciprofloxacin while retaining susceptibility to four or more non-β-lactam antibiotics. Moreover, this clone harbored a greater number of superantigenic enterotoxin genes (*seg*, *sei*, *sem*, *sen*, *seo*) compared with ST239, the previously dominant sequence type, which typically carried only a single enterotoxin gene (*sea*) [127]. Genomic characterization of the prevalent ST22-IV in Malaysia revealed that this clonal complex was both highly virulent and resistant, factors likely contributing to its widespread dissemination and eventual predominance over ST239 [125,127].

Although ST22 has exhibited widespread dissemination in Asia during certain periods, the present study observed a decline in its prevalence within clinical settings across the continent, coinciding with an increase in ST239.

Another sequence type identified in this study with pandemic potential is ST239. This clonal complex, one of the earliest recognized, is canonically associated with HA-MRSA [49,127]. Despite being reported in multiple countries, ST239 has never achieved widespread dissemination. In our study, it was primarily detected in clinical sequences from the Asian continent, ranking as the seventh most frequent sequence type (Figure 6, Table 1).

The significance of the ST239 clone emerged in the 1970s when it was first isolated from various regions worldwide [128]. Since then, the incidence of hospital-acquired and community-associated infections caused by ST239-MRSA has steadily increased over the decades [128]. ST239 is among the pandemic-potential clones that have persisted globally, having been isolated in Egypt [125], and across Asia, Europe, and the Middle East—including Iran [125,127,129]—as well as in both North and South America [128].

From a genetic perspective, ST239-MRSA is particularly notable, as it harbors six of the seven MLST maintenance genes with ST8 [130]. However, ST239 and its closest clonal relatives, ST240 and ST241, differ in the *arc*C allele, underscoring a genetic correlation between the ST8 and ST239 complexes [130]. Additionally, ST239 possesses the capacity to produce several toxins, primarily staphylococcal enterotoxins (SEs) A, K, and Q (*sea, sek,* and *seq*), along with lower levels of toxic shock syndrome toxin (TSST), exfoliative toxins A and B (ETA and ETB), and PVL [130]. These toxins contribute to significant pathogenic effects, posing substantial risks to the host [127,129].

The pandemic potential and molecular characteristics of ST239, including its capacity to produce enterotoxins, warrant closer surveillance. This is particularly relevant given its association with both hospital- and community-acquired infections, as well as its zoonotic implications, having also been linked to mastitis in cattle.

In environmental samples, the most frequently detected STs were ST398 and ST59 [123,130,131]. ST398 is prevalent in Asia and represents a major cause of infections in pigs, whereas ST59 is commonly associated with community-acquired infections in the region [124,131,132]. Approximately 47% of the environmental sequences analyzed in this study originated from Asia (Table 1), which likely accounts for the predominance of these STs in the environmental sample group.

To fully elucidate the significance of these clonal complexes, it is crucial to examine their distinct characteristics. ST398 first emerged among cattle herds in the early 2000s and subsequently began colonizing and infecting humans. Human MRSA infections have been classified into three primary categories based on their presumed sources: HA-MRSA, CA-MRSA, and community-onset healthcare-associated MRSA. More recently, a fourth category, LA-MRSA, was introduced to describe human MRSA infections linked to livestock exposure [133,134].

Notably, methicillin-resistant ST398 MRSA (LA-MRSA) has been consistently associated with livestock contact, whereas methicillin-susceptible ST398 MSSA has not. Genomic analyses indicate that ST398 MSSA is more frequently associated with humans, and key genetic differences exist between ST398 MRSA and ST398 MSSA, including the variable presence of genes conferring resistance to tetracycline and erythromycin [85,133]. Human colonization with the LA-MRSA ST398 was first identified among pig farmers in France and the Netherlands in the early 2000s. Since these initial reports, ST398 has been detected in various livestock species across multiple countries worldwide [133,134]. In Europe, ST398 MSSA is mostly associated with pigs, suggesting a potential transfer from human handlers to animals. Another critical observation is that ST398 LA-MRSA likely evolved from ST398 MSSA following the acquisition of antibiotic resistance genes and the loss of genes associated with human immune evasion [133]. Considering these findings, it can be concluded that ST398 LA-MRSA infections in humans represent a concerning zoonotic transmission back to the original host. This is particularly alarming given that this clonal complex has evolved from a methicillin-susceptible lineage into a resistant one, posing significant public health challenges.

Regarding ST59, although this sequence type has never been dominant in North America, it was first reported in San Francisco in the early 2000s as a community-associated strain. It was classified as PFGE type USA1000, carried the SCC*mec* type IV, and was predominantly negative for the PVL gene [135,136]. Shortly thereafter, reports of ST59 isolates began to emerge in Asia, initially occurring in Taiwan and later spreading to mainland China. Notably, the Asian clone carried the PVL gene and a novel *mec* element, later characterized as SCC*mec* type Vt [137,138].

In the present study, this sequence type was identified in environmental samples from the Asian continent and was the most frequently detected among environmental sequences, accounting for approximately 9.3% of cases (Figure 6 and Table 1). These findings align with previous reports in the scientific literature, as ST59 is widely isolated in community settings across Asia, where it is a leading cause of CA-MRSA infections [139]. Another significant aspect is that Asian ST59 isolates often harbor genes encoding toxins, including PVL, a virulence factor that targets and destroys leukocytes, thereby exacerbating the severity of skin and soft tissue infections [49,117]. Additionally, other virulence factors, such as hemolysins and superantigens, may also be present in ST59 isolates [140]. Collectively, these findings underscore the need for continued monitoring of this clonal complex.

The origins of ST59 remain a subject of debate. Some studies suggest that it originated in North America, while others propose an Asian origin [131]. What is evident, however, is that the Asian clone has evolved into two primary lineages: the Taiwan clone and the Asia-Pacific clone. The Taiwan clone, which carries the Sa2 phage encoding PVL, is associated with severe infections, whereas the Asia-Pacific clone, typically a commensal strain, carries the Sa3 phage encoding staphylokinase [130,138].

Beyond SCC*mec*, certain subclades of the Asian clones harbor a specific MGE known as the mobile element structure (MES) [139]. The MES is responsible for multidrug resistance, containing genes that confer resistance to erythromycin, clindamycin, kanamycin, streptomycin, and chloramphenicol [139]. The acquisition of MES is believed to have been a pivotal event enabling the widespread expansion of ST59 in East Asia, with regional variants of this element contributing to its diversification [84,138]. Alarmingly, in 2019, an outbreak of ST59 MRSA was reported in a neonatal intensive care unit in Hong Kong [141], highlighting this clonal complex’s potential to cause severe infections and reinforcing the urgent need for enhanced surveillance and control measures.

### 3.7. Associations Between SCCmec Types and MLST Profiles

The epidemiological profile of *Staphylococcus aureus* has undergone significant evolution over time, primarily owing to its remarkable ability to acquire resistance to multiple antibiotic classes. A crucial determinant of this resistance is the SCC*mec* element, specifically the *mec*A gene, which has been extensively identified across various *Staphylococcus* species [142]. This widespread distribution led to the hypothesis that *mec*A is situated on a mobile genetic element capable of mediating its transfer between organisms [4,7].

Subsequent studies substantiated that the emergence of methicillin-resistant staphylococcal strains resulted from the acquisition and incorporation of the *mec* element into the SCC*mec* of methicillin-susceptible strains [7,143]. Evidence suggests that the *mec*A gene originated in coagulase-negative Staphylococcus species and was later transferred to *S. aureus*, giving rise to MRSA [142]. The SCC*mec* element serves as the genetic vehicle facilitating this transfer, as *mec*A has not been detected in other staphylococcal species without an accompanying SCC*mec*-like structure [142,144].

To date, at least thirteen SCC*mec* types have been delineated, with certain types exhibiting a strong association with environments [144]. For instance, types I, II, and III are predominantly identified in strains isolated from hospital settings [142,145]. Notable HA-MRSA clones include CC5-SCC*mec* II (USA100), CC30-SCC*mec* II (EMRSA-16), and ST239-SCC*mec* III [142,144].

While all three SCC*mec* types (I, II, and III) carry the β-lactam resistance gene *mec*A, type III distinguishes itself by harboring additional antibiotic resistance genes within its J region. These genes include a transposon-encoding cadmium resistance determinants (ΨTn554) in the J2 region, as well as a tetracycline and mercury resistance plasmid (pT181) and another transposon (Tn554) conferring resistance to erythromycin and spectinomycin in the J3 region [144].

In the present investigation, SCC*mec* type III was identified in association with ST239 in samples from Asia (Figure 5), corroborating previous studies [52,125,127]. Notably, SCC*mec* type III is the largest among all SCC*mec* types, and the extensive genetic repertoire within its J region renders it particularly concerning in nosocomial environments, where such combinations may expedite the emergence and dissemination of multidrug resistance.

CA-MRSA clones exhibit distinct genetic profiles from HA-MRSA. The dominant clones include ST8-IV (USA300), ST1-IV (WA-1, USA400), ST30-IV (Southwest Pacific clone), ST59-V (Taiwan clone), and ST80-IV (European clone), with SCC*mec* types IV and V being the most frequently encountered [141,145]. In contrast, LA-MRSA clones, with ST398 being the most prominent representative, primarily harbor SCC*mec* types IV, V, and VII [142,146,147].

SCC*mec* type IV is particularly prevalent due to its adaptability to diverse epidemiological contexts, whereas the less common type VII has been identified in lineages adapted to specific environments, such as swine herds. In addition to ST398, other clonal complexes associated with LA-MRSA include ST9 and ST5, which share SCC*mec*-mediated resistance traits and are widely distributed among livestock populations in Asia and Europe [144,147].

The divergence in SCC*mec* type associations between CA-MRSA and LA-MRSA reflects distinct ecological adaptations. While community-associated clones exhibit greater genetic mobility and enhanced potential for dissemination in environments characterized by substantial human interaction, livestock-associated clones demonstrate high host specificity. Nevertheless, these clones are capable of transferring resistance traits to human lineages in occupational or environmental exposure scenarios.

The principal differences between SCC*mec* types are reflected in their size and genetic composition, which directly influence resistance acquisition [120,146,147,148]. For example, SCC*mec* type IV, the smallest identified to date, features a unique combination of a class B *mec* gene complex, a *ccr* type 2 gene, and the Tn4001 transposon within the J3 region [120,146]. Furthermore, SCC*mec* type IV has become one of the most frequently isolated types, primarily due to its prevalence in strains belonging to the rapidly expanding CA-MRSA group, such as strain CA05 (JCSC1968), isolated from joint fluid of a patient with septic arthritis and osteomyelitis, and strain 8/6-3P (JCSC1978), isolated from the perineum of another patient [131].

In the present study, SCC*mec* types IV and IVa, both linked to ST8, were the most frequently observed (Figure 5). Additionally, SCC*mec* type IIa, commonly detected in the dataset, was associated with ST5, aligning with previously reported findings. These results correspond to sequences from North America (Figure 5). In European sequences, ST239 is notably associated with SCC*mec* type IV and subtype IVh (Figure 5).

While SCC*mec* types I, II, and III were more prevalent in earlier years, SCC*mec* type IV has emerged as one of the most frequently isolated SCC*mec* types, largely due to its frequent occurrence in CA-MRSA strains, facilitating its rapid spread. SCC*mec* type IV is distinguished by its compact genetic structure, in contrast to the larger types I, II, and III, which enhances its mobility and enables its dissemination across various clonal complexes and environments, including both hospital and community settings. Although primarily associated with CA-MRSA, SCC*mec* type IV has also been identified in hospital isolates.

As previously noted, variations in the J region contribute to the diversity of SCC*mec* subtypes. Despite the smaller size of SCC*mec* type IV, its genetic composition within the J region exhibits considerable variability, resulting in numerous described subtypes (IVa to IVn). Subtypes IVa and IVh are particularly noteworthy for their structural differences within the J region, which impact additional antibiotic resistance profiles and the epidemiological characteristics of the strains they harbor.

Previous studies have highlighted the widespread presence of SCC*mec* type IV in clonal complexes adapted to various geographical and epidemiological contexts, such as ST8-SCC*mec* IV (associated with the USA300 clone in North America) and ST239-SCC*mec* IV in European and Asian regions [124,134,145]. The identification of subtype IVh in European isolates, as observed in this study, may reflect regional variations in selective pressures or gene flow among lineages.

In conclusion, the pathogenic dynamics of *S. aureus* are in a continual state of flux. SCC*mec* type IV, in particular, plays a critical role in this evolution due to its elevated genetic mobility and adaptability to diverse epidemiological contexts. Its widespread distribution, now evident in both traditional hospital strains and those from community settings, underscores its importance in the transmission of antimicrobial resistance and the increasing complexity of managing *S. aureus* infections.

### 3.8. Perspectives

MRSA remains a critical global health threat due to its substantial burden and enduring prevalence. MRSA is one of the foremost causes of both healthcare-associated and community-acquired infections worldwide. According to the Global Burden of Disease study, in high-income countries, approximately 50% of deaths attributed to AMR are linked to *S. aureus* and *E. coli*. The morbidity, mortality, and healthcare costs associated with MRSA underscore its critical status as a public health issue. Effectively addressing the challenges posed by MRSA necessitates a multifaceted strategy, encompassing sustained investments in research and development (R&D), enhanced infection prevention and control (IPC) measures, antimicrobial stewardship programs, and robust global surveillance efforts [149].

The use of antimicrobials in large-scale animal protein production surpasses human consumption by a factor of three. However, the emergence of antimicrobial resistance in microorganisms isolated from animals has garnered far less attention than resistance in human-associated pathogens [65,68,73,114,149]. This discrepancy is further emphasized by the considerable disparity in the number of *S. aureus* sequences derived from human versus environmental isolates in the NCBI database, highlighting the neglect of environmental samples in AMR research.

The 2024 WHO Bacterial Priority Pathogens List (BPPL) prioritization process embraced the One Health framework, recognizing the interdependence of human, animal, and environmental health [150]. Transmissibility across these sectors was a pivotal criterion in evaluating pathogens with outbreak potential or the capacity to disseminate resistance. Integrating the BPPL within One Health AMR policy frameworks could offer valuable direction for surveillance, research, and intervention strategies. This holistic approach is crucial for systematically monitoring and mitigating the advance of antimicrobial resistance across all sectors. Furthermore, addressing the existing knowledge gaps concerning the dynamics of resistance transmission within the One Health continuum is essential, particularly given the limited funding and insufficient evidence on the preventability and transmissibility of resistance in environmental and animal contexts.

The pressing need for a One Health approach becomes even more evident in light of these gaps. Systematic surveillance of antimicrobial resistance across the animal, environmental, and human sectors is imperative to prevent further exacerbation of the global AMR crisis.

### 3.9. Study Limitations

We acknowledge the significant underrepresentation of environmental sequences in comparison with clinical isolates, which impedes accurate monitoring of resistance genes and hinders the observation of *S. aureus* pathogenic evolution, considering its remarkable ability to disseminate and adapt across diverse environments. Another limiting factor was the lack of standardization in the metadata available for each sequence in the NCBI (National Center for Biotechnology Information) database. The absence of complete epidemiological data, or the presence of only partial information, resulted in the exclusion of more than 50% of the obtained sequences. A more comprehensive understanding of *S. aureus* pathogenicity could have been achieved had more complete data been available.

## 4. Materials and Methods

### 4.1. Genome Acquisition

*Staphylococcus aureus* genomes were retrieved from the NCBI Pathogen Detection website (https://www.ncbi.nlm.nih.gov/pathogens/, (accessed on 16 March 2023), hosted by the National Center for Biotechnology Information (NCBI). The genomes were accessed and downloaded on 16 March 2023, encompassing all genomes available at that time on the NCBI platform.

### 4.2. Screening of Metadata Belonging to Genomic Sequences

Using the command line interface, genomic sequences containing metadata (i.e., epidemiological data) were separated from those lacking such information. Sequences lacking epidemiological data were excluded from the analysis. Additionally, the N50 value, a statistical measure assessing the size distribution of the assembled contigs, was employed as a quality control criterion for each assembly. Sequences with lower values were discarded.

### 4.3. Characterization of the Presence and Absence of Genes

A custom database containing 140 genes associated with *S. aureus* biofilm production was compiled based on the scientific literature (Appendix A). The sequences of these genes were retrieved from the Virulence Finder database (https://www.mgc.ac.cn/cgi-bin/VFs/compvfs.cgi?Genus=Staphylococcus), (accessed on 20 March 2023). Once constructed, the database was employed with the ABRicate software (Software Version 1.0.1, released in 2023), (developed by Seemann, available on GitHub: https://github.com/tseemann/abricate, accessed on 20 March 2023) to identify the presence of these genes in the downloaded genomic sequences. A local bulk alignment of the sequences was performed for each genome, aligning them against the database using a threshold of 95% identity and 60% coverage. To ensure meaningful statistical analysis, frequency thresholds were set for the occurrence of each gene. Genes present in human clinical sequences with frequencies below 4000 were excluded from further analysis, while genes found in environmental sequences with frequencies lower than 50 were discarded.

### 4.4. Typing the Genomic Sequences

Genomic sequences were typed using the STARAMR software (Software Version 0.8.0, released in 2023), (https://github.com/phac-nml/staramr?tab=readme-ov-file#mlsttsv, accessed on 25 March 2023) to determine their clonal complexes. Multilocus sequence typing (MLST) evaluates a set of key genes that correlate with specific clonal complexes. STARAMR uses the PubMLST database (https://pubmlst.org/, accessed on 25 March 2023), which houses previously characterized genes for each clonal complex. The results of the MLST analysis, including sequence type (ST) and corresponding loci/alleles, were displayed with one genome per line. ABRicate software (Software Version 1.0.1, released in 2023) was also used to type the SCC*mec* elements and their subtypes by aligning the sequenced genomes against a database containing genomic island sequences.

Additionally, antimicrobial resistance (AMR) data from the metadata associated with each sequence was analyzed to assess the presence of resistance genes. Pathogen Detection software (Software Version 1.0, released in 2023) (https://github.com/dias-joaovictor/pathogendetection, accessed on 25 April 2023) was employed as a computational tool to input BioSample and AMR data via a CSV file. The software returned the presence (“TRUE”) or absence (“FALSE”) of genes, which were further processed for statistical analysis. The same frequency cutoff values were applied to analyze the resistance gene data.

### 4.5. Statistical Analysis

For statistical analysis, the CSV file containing gene presence or absence (“FALSE” or “TRUE”) was pre-processed. Entries marked “FALSE” (absence) were replaced with 0, and “TRUE” (presence) entries were converted to 1. This allowed for a quantitative analysis. The R programming language via the RStudio (Software Version 4.3.3, released in 2024) interface was used to conduct the following analyses:Calculation of gene frequencies.Evaluation of genetic correlations.Identification of genes with statistically significant associations.Analysis of the global distribution of MLST and SCC*mec* types.

This statistical workflow ensured accurate and reproducible insights.

### 4.6. Samples

#### Clinical and Environmental

The distinction between human clinical and environmental sequences was established as follows: Sequences derived from humans, regardless of their association with infections, along with those from hospital fomites, were categorized as human clinical sequences. Conversely, environmental sequences encompassed isolates from animals, irrespective of infection status, as well as those from food sources, terrestrial and aquatic environments, and other ecological settings.

## 5. Conclusions

In conclusion, this study identified key factors such as biofilm production and resistance genes that suggest potential co-dissemination, reinforcing the notion that the evolution of *S. aureus* clonal complexes—already recognized for their extreme virulence and resistance—is highly dynamic. Another significant finding was the bias in the NCBI public databases, which predominantly reflect *S. aureus* in the clinical contexts of wealthier countries and continents. The results further demonstrated that certain clonal complexes, including ST8, ST59, and ST398, exhibit considerable genetic diversity and adaptability, which likely contribute to their prevalence in both hospital and community settings. Additionally, environmental *S. aureus* isolates from regions with limited surveillance were underrepresented in the databases, highlighting a critical gap in global AMR data. Furthermore, genes such as *sasG* (containing the LPXTG motif), genes from the *isd* group, and those within the *ica*ADBC locus—all associated with biofilm production—showed widespread dissemination across both environmental and clinical isolates. Notably, the *sasG* gene emerged as the only biofilm-associated gene displaying a significant correlation with resistance genes, particularly with SCC*mec* type II. This observation further underscores the complex interplay between virulence and antimicrobial resistance, emphasizing the dual threat posed by such genetic elements. Mobile genetic elements, such as SCC*mec*, are critical to the rapid evolution and adaptation of *S. aureus*, enabling the acquisition and dissemination of both resistance and virulence determinants. This highlights the need for ongoing surveillance of these elements to better understand their role in the epidemiology and pathogenicity of *S. aureus*, aligning with previous literature emphasizing the diversity in Staphylococcus biofilm expression. Another important finding of this study was the association of resistance genes, including *fexA-B* (conferring resistance to florfenicol and chloramphenicol), *fusA* (resistance to rifampicin and fusidic acid), *aac(6′)-Ie/aph(2″)-Ia* (resistance to aminoglycosides), and *tet* (resistance to tetracycline). These genes were frequently observed together, showing notable linkage and distribution among the sequences, underscoring the critical need to monitor multiresistant clones. However, these findings warrant further investigation, particularly with respect to expanding database coverage and enhancing the understanding of the mechanisms behind biofilm formation and genetic co-dissemination in diverse environments.

## Figures and Tables

**Figure 1 antibiotics-14-00364-f001:**
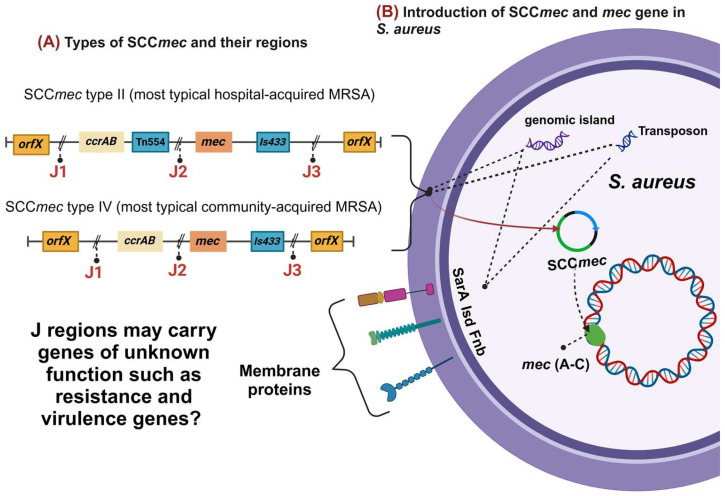
Development of resistance and distribution of biofilm-linked genes. (**A**)—Types of the Staphylococcal cassette chromosome *mec* (SCC*mec*) and its regions. The primary composition includes: the *orfX* gene, an open reading frame of unknown function that is highly conserved among clinical *S. aureus* strains; the *ccrAB* gene, which encodes a recombinase; the *Tn554* gene, a transposon; the *Is433* gene, an insertion sequence; the *mec* gene, which encodes β-lactamase; and the J regions (J1–J3), which are non-essential components of the chromosomal cassette. These J regions also facilitate the integration of transposons and insertion sequences, potentially carrying various genes. (**B**)—SCC*mec* and the *mec* region within the *S. aureus* chromosome. Genomic islands and transposons within the cell can integrate into SCC*mec*. These genomic elements harbor genes associated with the production of proteins that contribute to biofilm formation. Created in BioRender. De jesus, A. (2025), https://BioRender.com/l53e745 (accessed on 11 February 2025).

**Figure 2 antibiotics-14-00364-f002:**
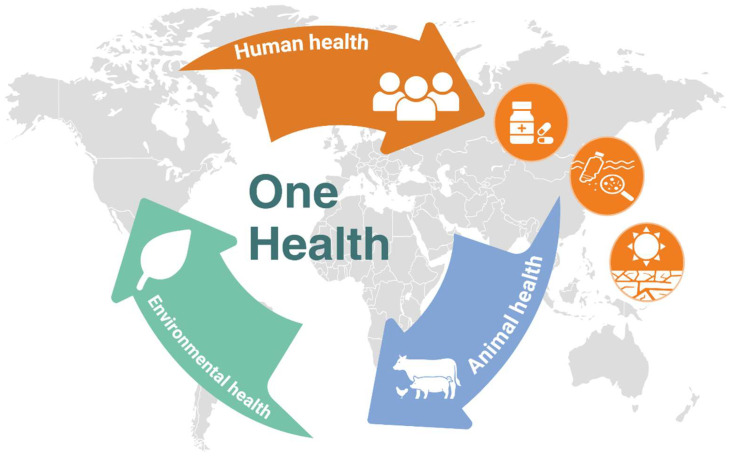
The One Health approach, based on the connection and observation between factors that influence human, animal, and environmental health. Within this paradigm, pollution, environmental degradation, and the overuse of pharmaceuticals—actions predominantly driven by human activity—have direct repercussions on the health of all entities involved. As illustrated, it is no longer possible to assess any risk to human health without taking animal and environmental health into account. Created in BioRender. De jesus, A. (2025), https://BioRender.com/h13b188 (accessed on 11 February 2025).

**Figure 3 antibiotics-14-00364-f003:**
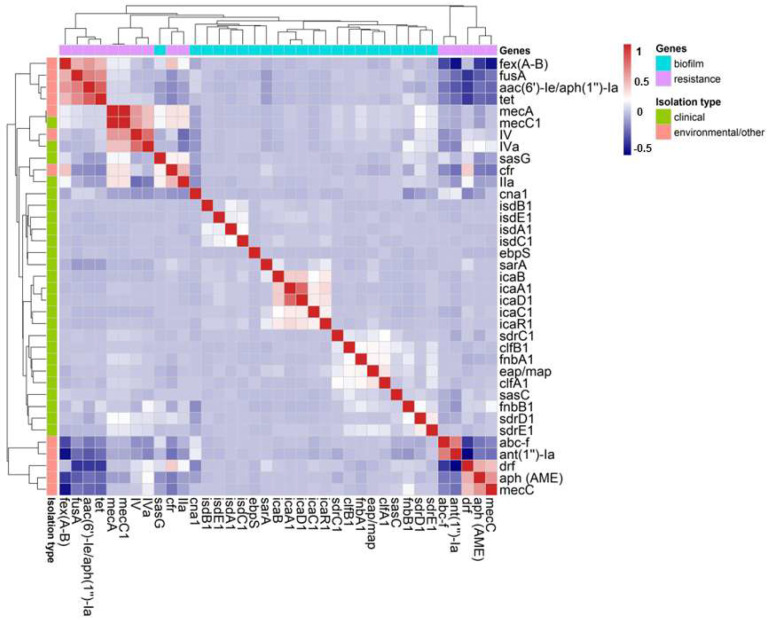
Heatmap showing the correlation of genes identified in the analyzed sequences. In the heatmap, gene correlations are visually represented using a color gradient from red to blue, corresponding to values ranging from 1 to −5, respectively. Additionally, two distinct clusters are displayed: one categorizing isolate type, where clinical isolates are marked in green and environmental isolates in pink, linked to the adjacent dendrogram. At the top of the heatmap, a second dendrogram organizes genes into two primary groups—those associated with biofilm production (depicted in blue-green tones) and those linked to antimicrobial resistance mechanisms (shown in purple).

**Figure 4 antibiotics-14-00364-f004:**
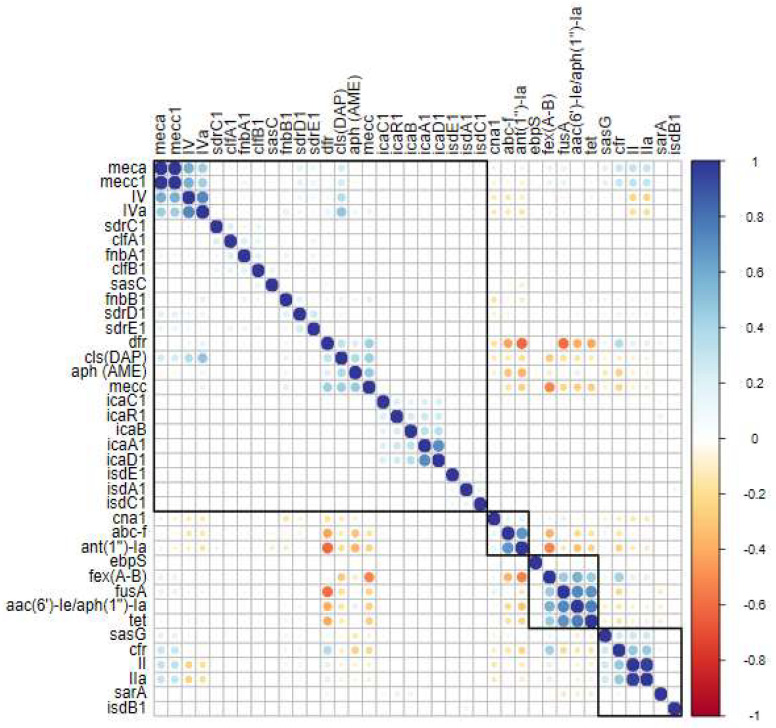
Correlation matrix of genes in human clinical sequences. This heatmap visualizes the correlation matrix of genes identified in sequences classified as human clinical. Genes exhibiting the highest correlation are marked in blue, with varying shades representing values from 0 to 1, indicating stronger associations. Conversely, less related genes appear in orange, with shades ranging from yellow to red, representing values from 0 to −1. Clusters of highly correlated genes are highlighted with black squares. Notably, the analysis reveals a distinct block of genes associated with SCC*mec* type IV and IVa, among others, emphasizing their strong correlation within human clinical sequences.

**Figure 5 antibiotics-14-00364-f005:**
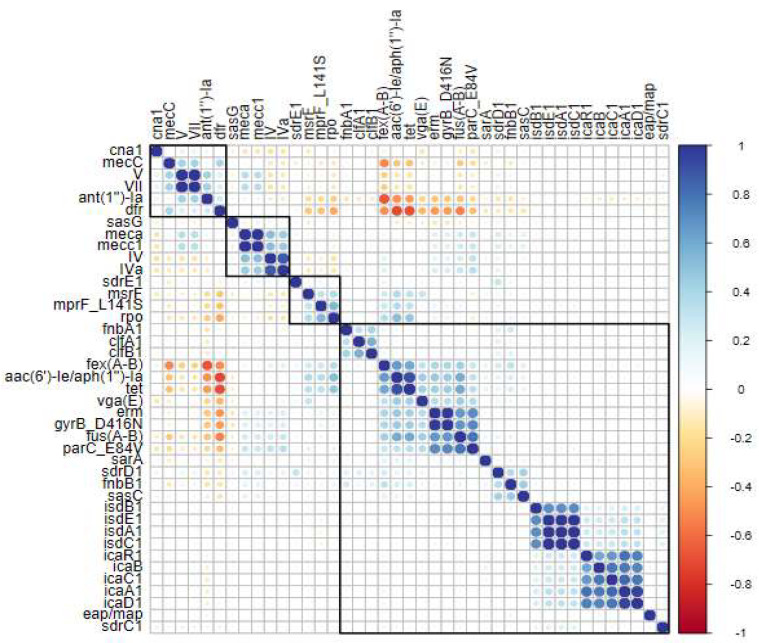
Correlation matrix of genes in environmental sequences. This heatmap visualizes the correlation matrix of genes identified in sequences classified as environmental. Genes with the highest correlation are marked in blue, with shades representing values from 0 to 1, indicating stronger associations. Conversely, less related genes appear in orange, with yellow to red tones representing values from 0 to −1. Clusters of highly correlated genes are highlighted with black squares. Notably, this correlation analysis reveals a greater number of related genes and a broader diversity of resistance genes present in environmental samples.

**Figure 6 antibiotics-14-00364-f006:**
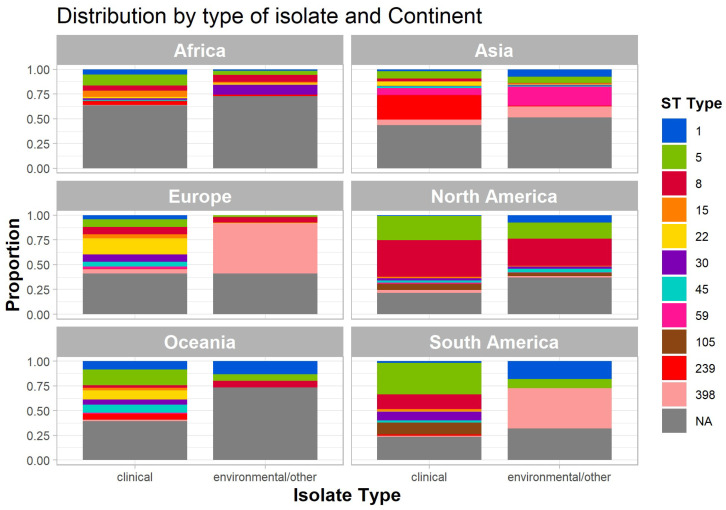
Global distribution of sequence typing (MLST). The graph illustrates the distribution of sequence types (STs) across continents for each type of isolate (clinical or environmental). Each ST is represented by a distinct color in the stacked bars for each isolate type. Additionally, ‘NA’ indicates unidentified STs.

**Figure 7 antibiotics-14-00364-f007:**
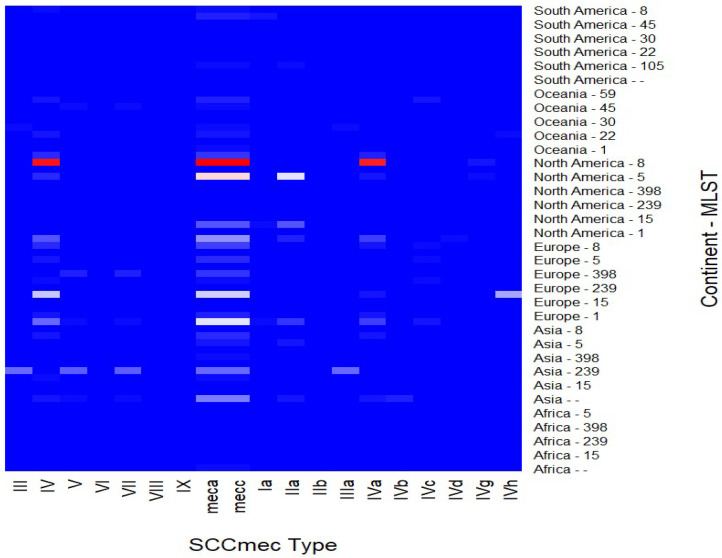
Heatmap of *SCCmec* types distribution across MLSTs and continents. The heatmap illustrates the prevalence of SCC*mec* elements among multilocus sequence types (MLSTs) in human clinical and environmental isolates across different continents. The intensity of the red color represents the relative frequency, with variations reflecting regional and genotypic patterns. A gradual transition from white to red indicates higher frequencies.

**Table 1 antibiotics-14-00364-t001:** Absolute frequency of sequences by continent and MLST (multilocus sequence typing) of *S. aureus*.

Variable	N	Clinical, N = 42,188 ^1^	Environment/Others, N = 1881 ^1^	*p*-Value ^2^
Continents	44,069			<0.001
Africa		857 (2.0%)	70 (3.7%)	
Asia		4674 (11%)	889 (47%)	
Europe		15,957 (38%)	408 (22%)	
North America		15,832 (38%)	477 (25%)	
Oceania		3496 (8.3%)	15 (0.8%)	
South America		1372 (3.3%)	22 (1.2%)	
MLST	44,069			<0.001
-		14,239 (34%)	870 (46%)	
8		7501 (18%)	162 (8.6%)	
5		6504 (15%)	146 (7.8%)	
22		3120 (7.4%)	11 (0.6%)	
30		1910 (4.5%)	25 (1.3%)	
398		1527 (3.6%)	321 (17%)	
45		1535 (3.6%)	26 (1.4%)	
239		1472 (3.5%)	8 (0.4%)	
1		1269 (3.0%)	110 (5.8%)	
105		1182 (2.8%)	15 (0.8%)	
15		1090 (2.6%)	13 (0.7%)	
59		839 (2.0%)	174 (9.3%)	

^1^ n (%). ^2^ Pearson’s chi-square test.

**Table 2 antibiotics-14-00364-t002:** Absolute frequency of genes in global sequences of *S. aureus*.

**Isolation Type**	**Number of Sequences**
Clinical	42,188 (96%)
Environmental/Other	1881 (4.3%)
Total	44,069
**Genes**	**Sequence Frequency**
*eap/map*	43,833 (99%)
*ebps*	44,059 (100%)
*sarA*	41,760 (95%)
*sasG*	5635 (13%)
*clfA1*	43,533 (99%)
*clfB1*	43,906 (100%)
*cna1*	8557 (19%)
*fnbA2*	43,207 (98%)
*fnbB1*	35,666 (81%)
*icaA2*	44,013 (100%)
*icaB*	43,854 (100%)
*icaC1*	43,834 (99%)
*icaD1*	44,005 (100%)
*icaR2*	43,857 (100%)
*isdA1*	44,059 (100%)
*isdB1*	44,032 (100%)
*isdC1*	44,060 (100%)
*isdE1*	44,052 (100%)
*sasC*	43,508 (99%)
*sdrC1*	43,317 (98%)
*sdrD1*	38,972 (88%)
*sdrE1*	40,718 (92%)
**ARG**	**Sequence Frequency**
*aac(6′)-Ie/aph(1″)-Ia*	9694 (22%)
*abc-f*	6518 (15%)
*ant(1″)-Ia*	11,877 (27%)
*aph (AME)*	12,459 (28%)
*cfr*	6435 (15%)
*dfr*	21,895 (50%)
*fexA-B*	20,051 (45%)
*fusA*	12,826 (29%)
*mecC*	15,268 (35%)
*tet*	6786 (15%)
*IV*	14,065 (32%)
*mecA*	25,657 (58%)
*mecC2*	25,733 (58%)
*IIa*	5500 (12%)
*IVa*	9537 (22%)

ARG: Antimicrobial resistance genes.

## Data Availability

The original contributions presented in this study are included in the article/Appendix A. Further inquiries can be directed to the corresponding author. Whole genome sequence data are available in GenBank, with all genomes collected on 16 March 2023. Further gene and database information is provided in the Appendix A.

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
