# Peer review of "Distribution of Antimicrobial Resistance and Biofilm Production Genes in the Genomic Sequences of S. aureus: A Global In Silico Analysis"

_antibiotics, 2025, doi:10.3390/antibiotics14040364_

Round 1
Reviewer 1 Report
Comments and Suggestions for Authors
- How did you ensure that the dataset from NCBI does not introduce selection bias (e.g., overrepresentation of certain regions)?
- Were there any steps taken to validate the gene annotation pipeline? Can you provide more details on the thresholds for gene presence/absence?
- Why was a 95% identity and 60% coverage threshold chosen for gene identification? Was this based on a previous study?
- The Pearson’s chi-square test is mentioned for categorical data. Were any multiple testing corrections applied?
- Did you account for phylogenetic relationships between strains when interpreting resistance gene associations?
- Can you clarify if the correlation between biofilm and AMR genes is a true functional relationship or just a co-occurrence?
- Is there evidence from functional genomics or transcriptomics that sasG, mecA, and cfr interact biologically?
- What steps were taken to correct for underrepresented regions (e.g., Africa & South America)?
- Could there be technical or publication biases that skew SCCmec distributions by region?
- How confident are you in SCCmec subtyping from short-read assemblies?
- Was long-read sequencing considered to validate ambiguous SCCmec structures?
Author Response
Dear Reviewer,
We sincerely appreciate your valuable contributions to this work. We have carefully considered and incorporated all your suggestions and have addressed each of the questions raised. Please do not hesitate to contact us if further clarification is required.
Thank you once again for your insightful feedback.
Best regards,
Ana Carolina Silva de Jesus
1- How did you ensure that the dataset from NCBI does not introduce selection bias (e.g., overrepresentation of certain regions)?
We have already taken the overrepresentation of certain regions into account. Unfortunately, sequencing deposits in the NCBI of strains originated from human, animal, and environmental health are randomly worldwide. Some countries do not have such data available in NCBI, while others have very few genomes available. As expected, North America and Europe lead in sequencing efforts and the availability of data in NCBI. Therefore, we have explicitly stated throughout the document that the data are more representative of these regions. In addition, the lack of uniformity in epidemiological data also contributed to this overrepresentation. We avoided the download of 50,531 genomes (53.4% of the data) due to the missing of metadata information, and we knew that this loss would impact the dataset, increasing the overrepresentation bias.
2- Were there any steps taken to validate the gene annotation pipeline? Can you provide more details on the thresholds for gene presence/absence?
We believe there is no need to validate the gene annotation process because the genes are identified through BLAST alignment against resistance genes contained in the curated ResFinder database using the STARAMR and Abricate software. We have taken care to use the most up-to-date databases. The ResFinder database is continuously curated and updated, being widely used and accepted by the scientific community, which eliminates the need for additional validation.
Regarding the thresholds for gene presence/absence, we defined a gene as "present" if it met a minimum identity threshold of 95% and a coverage threshold of 60%. These parameters were selected to balance sensitivity and specificity, allowing the detection of fragmented or truncated genes while minimizing false positives. Genes that did not meet these thresholds were considered "absent," unless corroborated by alternative detection methods. Additionally, we evaluated sequence coverage distributions across genomes to ensure that our cutoffs were biologically meaningful and aligned with previous studies in the field.
3- Why was a 95% identity and 60% coverage threshold chosen for gene identification? Was this based on a previous study?
The 95% identity and 60% coverage thresholds were chosen to balance the inclusion of true gene variants while minimizing false positives from distantly related sequences. In exploratory analyses, as was our case, the thresholds can be adjusted to capture a greater diversity of sequences. However, the 95% identity value is high enough to ensure that the identified sequence is truly homologous to the gene of interest and not a similar gene from another species or a sequence distorted by assembly errors. The 60% coverage value allows the detection of genes that may be fragmented or slightly truncated. These also avoid that genes lying on the edge of a contig or spread over two contigs were not missed due to possible non-perfect assembly. Additionally, we based our approach on previous studies from our group, as well as other exploratory works that utilized the same identity and coverage thresholds.
4- The Pearson’s chi-square test is mentioned for categorical data. Were any multiple testing corrections applied?
Yes, we applied the Benjamini-Hochberg correction to maintain statistical power. Using the tbl_summary() function in R, we applied add_p() to compare groups and generate p-values.
5- Did you account for phylogenetic relationships between strains when interpreting resistance gene associations?
We did not incorporate phylogenetic relationships between strains when interpreting resistance gene associations. Our study focused on analyzing assembled genomes in contigs, identifying resistance genes in Staphylococcus genomes, and determining the SCCmec type for each strain to associate with resistance profiles.
In studies like ours, which rely on publicly available genomes from databases such as GenBank, each isolate is typically considered unique. These genomes are usually derived from single bacterial colonies rather than representing multiple isolates from the same clonal lineage. As a result, we did not assess the phylogenetic similarity between isolates to determine levels of clonality that could potentially bias the observed associations. Unlike outbreak investigations or evolutionary studies where phylogenetic context is crucial, our approach follows the standard methodology for prevalence studies based on genome databases, where each entry is treated as an independent observation.
6- Can you clarify if the correlation between biofilm and AMR genes is a true functional relationship or just a co-occurrence?
In our study, we can only demonstrate the co-occurrence of genes associated with both biofilm formation and antimicrobial resistance (AMR). It would not be prudent to state a true functional relationship. However, there are studies in the scientific literature indicating a possible functional relationship between biofilm formation and AMR gene transfer. These studies have found that Ccr recombinases mediate the excision and insertion of SCC into the attB locus and that some genes may be more likely to be incorporated and disseminated within a biofilm.
7- Is there evidence from functional genomics or transcriptomics that sasG, mecA, and cfr interact biologically?
There is no direct or well-established evidence that the sasG, mecA, and cfr genes interact biologically in a functional or regulatory manner. However, there are some indirect connections that may suggest interactions in the context of antimicrobial resistance and bacterial adhesion in Staphylococcus spp. The sasG gene facilitates biofilm formation, and it is known that bacteria in biofilms may have increased expression of resistance genes, including mecA and cfr. Biofilms create a protected environment that reduces the efficacy of antibiotics and may select for the persistence of resistant cells. Furthermore, studies indicate that mecA and cfr may be present simultaneously in multidrug-resistant Staphylococcus aureus. The SCCmec cassette, which carries mecA, may occasionally contain other resistance genes, and there are reports of cfr-containing plasmids in MRSA (methicillin-resistant) strains.
8- What steps were taken to correct for underrepresented regions (e.g., Africa & South America)?
Since our study explores the frequency of these genes, we believe it is important to present the data despite the underrepresentation. To address this issue, we combined sequences from different countries and distinguished, for example, South America from North America, knowing that South America would be underrepresented. This approach allows us to emphasize two key points: (1) The urgent need for increased surveillance in underrepresented regions, as antimicrobial resistance monitoring requires a global effort, and (2) The significant differences in sequence types (ST) between continents, which can impact the genetic repertoire of strains and influence outbreak control strategies.
9- Could there be technical or publication biases that skew SCCmec distributions by region?
Yes, technical and publication biases can affect the reported distributions of SCCmec types by region. These biases can distort the perception of which variants are predominant in certain geographic areas. For example, SCCmec detection can be influenced by the method used. Sequencing can detect variants not identified by older methods, leading to the undercharacterization of certain SCCmec types. Less sensitive methods may also result in the underestimation of SCCmec types, particularly those with rearrangements or deletions. Additionally, many studies prioritize lineages that cause outbreaks or severe infections, leading to the overrepresentation of SCCmec types associated with hospital-related antimicrobial resistance, while rare SCCmec types or those less associated with virulence may be underreported. To mitigate these biases, we used the SCCmecFinder database (via ABRicate) to reanalyze SCCmec sequences and compare them with metadata. However, we acknowledge that some biases remain unavoidable.
10- How confident are you in SCCmec subtyping from short-read assemblies?
SCCmec subtyping using short-read sequencing is moderately reliable for well-characterized classical types but presents challenges for recombinant, incomplete, or novel variants. Ideally, short-read sequencing data should be combined with long-read sequencing to ensure greater typing accuracy and avoid errors due to fragmented assembly. Since we worked with NCBI-available data (i.e., assembled and published sequences) and focused on well-characterized classical SCCmec types, identification was generally reliable. Additionally, we used the SCCmecFinder tool to compare available metadata with our findings.
11- Was long-read sequencing considered to validate ambiguous SCCmec structures?
No, long-read sequencing data (PacBio or Oxford Nanopore) were not utilized in this study. We focused exclusively on classical SCCmec types and did not apply long-read sequencing to ambiguous structures. We acknowledge that long-read sequencing is increasingly used to validate ambiguous SCCmec structures. However, the primary objective of this work was to analyze well-characterized SCCmec types and their potential associations with biofilm-related and resistance genes. Therefore, we only used metadata and SCCmecFinder database results (via ABRicate) to validate the most classical SCCmec types.
Reviewer 2 Report
Comments and Suggestions for Authors
Review for Antibiotics: Title: Distribution of Antimicrobial Resistance and Biofilm Production Genes in the Genomic Sequences of S. aureus: A Global in-Silico Analysis
Authors: Ana Carolina Silva de Jesus, Rafaela G. Ferrari, Pedro Panzenhagen, Anamaria M. P. dos Santos, Ana Beatriz Portes, Carlos Adam Conte-Junior * Antibiotics Use and Antimicrobial Stewardship
Comments and Suggestions for Authors (will be shown to authors)
This is a very timely and important body of research. It reads more like an overview of MRSA population genetics and mobile genetic elements on a global scale which is useful. The authors have investigated over 44k S. aureus genomes. They have been very clear regarding any shortcomings regarding the study. For example, there were less genomes associated with environmental S. aureus compared to clinical samples. There was also variation between geographic regions which are due to resource and finance restraints.
The following are some comments and minor suggestions to consider. I am happy to recommend publication once these are addressed.
The authors limit their discussion to the main fundamental SCCmec elements as well as some of the other SCC elements that have been identified. They haven’t mentioned or discussed anything relating to the SCCmec and/or SCC-like remnants that have been investigated. This would be a potentially vast area to discuss of course. It may be worth highlighting the importance of this region in orfX where these SCC mobile genetic elements appear to be very important and appear to exhibit quite a bit of variation.
At several points, (lines 213-219, line 528) the authors discuss the selective bias towards investigating clinical isolates. This is an important discussion point and it should be a balanced point. Many research projects/groups will be able to justify or qualify this bias with the questions they are trying to answer or the locations they are working in, i.e. if a research group or reference lab are trying to investigate an outbreak in a hospital they may prioritize clinical samples even though significant environmental contamination has been shown previously and identical isolates have been detected from the patient, healthcare workers and the patients near environment.
I am curious if the authors had to exclude much of the genomes they found in the database due to poor quality? This has been flagged as an issue in the NCBI GenBank database.
The authors have also discussed the important point of the lack of metadata. This study would have benefitted if there was a standardised method for inclusion of metadata in online repositories for whole genome data.
Under section 2.6 Is it appropriate that isolates recovered from animals are deemed environmental? Should there be separate categories for animals and environmental?
Minor comments.
Line 62: I believe the function for the orfX is now known.
line 1101-1104: I am wondering if the authors are suggesting that the biofilm and resistance genes are driving forces for co-dissemination. Could the authors clarify this please.
Author Response
Dear Reviewer,
We sincerely appreciate your valuable contributions to this work. All of your suggestions have been carefully considered and incorporated, and we have addressed all the questions raised. Please do not hesitate to reach out if further clarification is needed.
Thank you once again for your insightful feedback.
Best regards,
Ana Carolina Silva de Jesus
Reviewer- “The authors limit their discussion to the main fundamental SCCmec elements as well as some of the other SCC elements that have been identified. They haven’t mentioned or discussed anything relating to the SCCmec and/or SCC-like remnants that have been investigated. This would be a potentially vast area to discuss of course. It may be worth highlighting the importance of this region in orfX where these SCC mobile genetic elements appear to be very important and appear to exhibit quite a bit of variation.”
Answer (Excerpt to be inserted in the article)
Dear Reviewer,
Thank you for your insightful comments. We have incorporated a discussion on SCCmec and SCC-like remnants in the introduction (lines 73-107), emphasizing their importance in the orfX region and their role in genetic plasticity and variation among S. aureus strains. This addition highlights the significance of these elements in pathogen evolution and antimicrobial resistance dynamics. We appreciate your valuable feedback, which has helped improve the clarity and comprehensiveness of our manuscript.
Reviewer- “At several points, (lines 213-219, line 528) the authors discuss the selective bias towards investigating clinical isolates. This is an important discussion point and it should be a balanced point. Many research projects/groups will be able to justify or qualify this bias with the questions they are trying to answer or the locations they are working in, i.e. if a research group or reference lab are trying to investigate an outbreak in a hospital they may prioritize clinical samples even though significant environmental contamination has been shown previously and identical isolates have been detected from the patient, healthcare workers and the patients near environment.”
2- Answer (Excerpt to be inserted in the article)
Dear Reviewer,
Thank you for your valuable feedback. We have expanded our discussion (Lines 470-484) to provide a more balanced perspective on the selective bias towards clinical isolates. Specifically, we acknowledge that many research groups and reference laboratories prioritize clinical samples based on the objectives of their studies, such as outbreak investigations in hospital settings. Additionally, we now emphasize that while clinical isolates are often the primary focus, significant environmental contamination and identical isolates from patients, healthcare workers, and hospital surfaces highlight the importance of integrating environmental surveillance into broader epidemiological studies.
We appreciate your insightful comments, which have helped refine and strengthen our discussion.
Reviewer- “I am curious if the authors had to exclude much of the genomes they found in the database due to poor quality? This has been flagged as an issue in the NCBI GenBank database.”
3- Answer
Dear Reviewer,
To ensure the reliability of our analysis, we applied strict quality control criteria for genome selection. Genomes with low sequencing coverage, incomplete assemblies, or poor annotation quality were excluded. For this purpose, the N50 statistic was used, which represents the median contig size, giving greater weight to larger contigs.
Additionally, we specifically removed sequences that contained excessive gaps, ambiguous metadata, or lacked essential metadata. This filtering process was conducted using Unix-based scripting, executed in a command-line environment, to identify sequences with complete metadata.
From an initial dataset of 94,600 sequences, a total of 50,531 genomes (53.4%) were excluded due to these quality constraints. The remaining 44,069 high-quality genomes were retained for further analysis, ensuring a robust and reliable dataset for investigating the distribution of SCCmec, resistance genes, and biofilm-associated elements.
I hope we have managed to clarify your question.
Reviewer- “The authors have also discussed the important point of the lack of metadata. This study would have benefitted if there was a standardised method for inclusion of metadata in online repositories for whole genome data.”
4- Answer
Dear Reviewer,
We appreciate your insightful comment regarding the lack of metadata in publicly available genome repositories. Indeed, the absence of standardized metadata significantly limits the depth of genomic analyses, particularly when investigating epidemiological patterns and strain origins.
In our study, we encountered substantial variability in the completeness and quality of metadata, which restricted certain comparative analyses. The implementation of a standardized system for metadata inclusion in whole-genome sequencing (WGS) repositories would greatly enhance the reliability and reproducibility of large-scale studies like ours. We acknowledge that more comprehensive metadata would have strengthened our findings, particularly in correlating genomic characteristics with clinical and environmental contexts.
This limitation underscores the need for international initiatives and guidelines to ensure consistent metadata annotation across genomic databases. Future efforts in genomic surveillance should prioritize improving metadata curation to facilitate more robust analyses of antimicrobial resistance and pathogen evolution.
Thank you once again for your valuable feedback
Reviewer- “Under section 2.6 Is it appropriate that isolates recovered from animals are deemed environmental? Should there be separate categories for animals and environmental?”
5- Answer
Dear Reviewer,
The decision to combine animal and environmental isolates was made due to their underrepresentation in the dataset. The total number of animal and environmental isolates available for analysis was only 1,881 sequences, compared to 42,188 clinical sequences. Additionally, some countries did not have any data on animal isolates, further limiting the dataset. For this reason, we merged these categories to ensure a more comprehensive analysis.
Furthermore, it did not seem appropriate to group animal isolates with human isolates, even though it is well established that most animal strains originate from human transmission. The main issue is that the majority of the human isolates sequences were obtained from infections, whereas animal isolates may not have originated from human infections and could already be colonizing the animals independently. However, the limited number of sequences was the predominant factor in our decision to merge animal and environmental isolates.
Thank you once again for your valuable feedback
Reviewer- “Line 62: I believe the function for the orfX is now known.”
6- Answer
Dear Reviewer, the section clarifying the function of orfX has already been updated (lines 76–82).
Reviewer- “line 1101-1104: I am wondering if the authors are suggesting that the biofilm and resistance genes are driving forces for co-dissemination. Could the authors clarify this please.”
7- Answer
Dear Reviewer,
The co-dissemination of biofilm-associated genes and antimicrobial resistance (AMR) genes in Staphylococcus aureus can be attributed to genetic linkage, selective pressure, and horizontal gene transfer mechanisms. Biofilm formation enhances bacterial persistence and survival in hostile environments, including those with antibiotic exposure, which in turn selects for resistant strains.
One key factor in this co-selection process is the presence of mobile genetic elements (MGEs), such as plasmids, transposons, and staphylococcal cassette chromosome (SCC) elements, which frequently carry both AMR genes and biofilm-related genes. For example, certain SCCmec types not only harbor mecA, conferring methicillin resistance, but also ica operons, which encode proteins essential for exopolysaccharide production in biofilms.
Additionally, biofilm structures protect bacterial communities from antibiotic penetration and host immune responses, increasing the likelihood of genetic exchange via conjugation, transformation, and bacteriophage-mediated transduction. This facilitates the co-transfer of resistance and biofilm-associated genes within bacterial populations.
Selective pressure exerted by antibiotic treatments in hospital environments further reinforces this process, as biofilm-forming strains are more likely to persist and acquire additional resistance determinants. As a result, strains that exhibit both strong biofilm formation and multidrug resistance have a greater advantage in colonization and infection, promoting their widespread dissemination.
These findings highlight the importance of understanding the genetic and environmental factors driving the co-evolution of biofilm formation and antimicrobial resistance, as they have significant implications for infection control strategies and therapeutic development.
I hope we have managed to clarify your question.
Round 2
Reviewer 1 Report
Comments and Suggestions for Authors
accept